# FIDELITY-AWARE DATA COMPOSITION FOR ROBUST ROBOT GENERALIZATION

## ABSTRACT

Generalist robot policies trained on large-scale, visually homogeneous datasets can be susceptible to shortcut learning, which impairs their out-of-distribution (OOD) generalization. While generative data augmentation is a common approach to introduce diversity, it presents a subtle challenge: data composition. Naively mixing real and synthetic data can corrupt the learning signal, as this process often prioritizes visual diversity at the expense of information fidelity. This paper suggests that robust generalization depends on principled, fidelity-aware data composition. We introduce Coherent Information Fidelity Tuning (CIFT), a framework that treats data composition as an optimization problem. CIFT uses a practical proxy for Information Fidelity based on the feature-space geometry of a dataset. This enables the identification of a phase transition, termed the Decoherence Point, where training stability degrades. The framework includes a generative engine, Multi-View Video Augmentation (MVAug), to synthesize a causally disentangled data spectrum for this tuning process. Applying CIFT to policy architectures such as $\pi_0$ and GE-Act improves OOD success rates by over 54%. The datasets used in this study are available in the anonymous repository provided. All model checkpoints will be released in a public repository after the review process to facilitate reproducibility. The anonymous code repository is available at: https://anonymous.4open.science/r/CIFT-code.

## 1 INTRODUCTION

Training large-scale, data-driven generalist policies is a central approach in modern robotics. Vision-Language-Action (VLA) models are a prominent example, which demonstrate the capacity for performing tasks in unstructured environments (Brohan et al., 2023; Black et al., 2025; Firoozi et al., 2025; O'Neill et al., 2024). The premise is that broad capabilities emerge when models learn statistical patterns from datasets that have high fidelity to the real world's causal structure.

However, this premise is often not met in practice. The significant cost and complexity of acquiring comprehensive real-world data lead to training sets with inherent statistical biases, for example, limited backgrounds, textures, and lighting. These biases can foster low-fidelity statistical cues, such as spurious correlations between an action and a background texture. This divergence between the correlations in the training data and the true causal relationships of a task creates a data-fidelity gap. This gap can drive policies toward shortcut learning (Geirhos et al., 2020), where they exploit these low-fidelity, "spurious" cues over more predictive (Ribeiro et al., 2016; Beery et al., 2018), "core" causal ones (Singla & Feizi, 2022; Hermann et al., 2024). The result is policies that generalize poorly and are prone to exhibit failures on specific subgroups of data where learned shortcuts become invalid, a known challenge for out-of-distribution (OOD) generalization (Sagawa et al., 2020).

A common strategy for the data-fidelity gap is to use generative models to create synthetic augmentations (Bowles et al., 2018). The goal is to increase visual diversity (e.g., by changing backgrounds or textures) to prevent policies from relying on spurious correlations. However, unprincipled data mixing can be counterproductive (Cubuk et al., 2019); it presents a trade-off where the diversity from synthetic data can come at the cost of the information fidelity of real demonstrations. (De Haan et al., 2019; Park et al., 2021) An excessive amount can dilute the original learning signal, leading to unstable training or a decline in performance. The central challenge is therefore not just the synthesis of varied data, but the principled composition of the final training dataset (Bansal et al., 2024).

Figure 1: The CIFT framework pipeline. Given a small seed dataset, our generative engine, MVAug, synthesizes a large pool of augmented data. CIFT then analyzes this pool to select a suitable data mixture that maintains information fidelity. The resulting curated dataset is used to train a robust policy that generalizes to novel environments.

This work proposes a method for systematic data composition, as overviewed in Figure 1. The proposed framework integrates a generative engine, Multi-View Video Augmentation (MVAug), with a composition algorithm, Coherent Information Fidelity Tuning (CIFT). CIFT determines a mixing ratio by analyzing learning dynamics, with the objective of improving generalization while maintaining performance on the original task distribution. Our main contributions are:

1. Multi-View Video Augmentation (MVAug): a video-to-video augmentation engine for synthesizing multi-view consistent, causally disentangled robotic demonstrations.

2. Coherent Information Fidelity Tuning (CIFT): a data composition framework guided by a proposed metric, Information Fidelity, to optimize the data mixing ratio and ensure training stability.

3. Extensive empirical validation: a demonstration that CIFT improves the OOD success rate of widely-used policies by over 54% by mitigating shortcut learning.

## 2 RELATED WORK

**Generalist Robot Policies.** Robotics research increasingly centers on training high-capacity, generalist policies on large-scale datasets (Reed et al., 2022; Walke et al., 2023; O'Neill et al., 2024; Khazatsky et al., 2024). This approach has led to the development of various architectures, from transformers (Zitkovich et al., 2023; Brohan et al., 2023; Driess et al., 2023) to vision-language models (Kim et al., 2024). The performance of this paradigm, however, is often constrained by data acquisition. The significant cost and complexity of collecting diverse real-world data can result in training sets that are visually homogeneous, a characteristic linked to the fragmentation of aggregated datasets (Dasari et al., 2020; Xing et al., 2025). This can create a data-fidelity gap, where the training distribution does not fully capture the causal structure of real-world environments (Chebotar et al., 2019). This gap is a contributing factor to poor out-of-distribution (OOD) generalization, especially on coherent data subgroups where spurious correlations fail (Sagawa et al., 2020).

**Shortcut Learning in Robotics.** Shortcut learning is a primary consequence of the data-fidelity gap, where models adopt decision rules that perform well on standard benchmarks but show poor generalization to new environments (Geirhos et al., 2020; Ye et al., 2024). Policies trained on biased data may learn to exploit spurious features (Baker et al., 2019; Izmailov et al., 2022; Singla & Feizi, 2022), such as background textures that are predictive in the training set (Xiao et al., 2021; Luo et al., 2021; Tobin et al., 2017). Such features are often learned because they are highly available, meaning they are easy for a model to extract. This reliance on spurious correlations is a known characteristic of deep nonlinear models, which can prioritize feature availability over causal predictivity (Hermann et al., 2024). Applying certain training paradigms, such as distributionally robust optimization (DRO), may be insufficient without careful regularization (Sagawa et al., 2020),

and some methods like adversarial or contrastive training may even increase background sensitivity (Moayeri et al., 2022). This issue is particularly relevant in robotics, where dataset fragmentation can foster the learning of shortcuts (Xing et al., 2025), presenting a barrier to deployment.

**Data Augmentation for Generalization.** To address the challenges of data scarcity and shortcut learning, data augmentation has become a widely used strategy. Recent work has advanced data *synthesis* for creating varied robotic demonstrations. This includes methods for background randomization (Chen et al., 2023; Teoh et al., 2024; Yuan et al., 2025), semantically conditioned modifications (Chen et al., 2024), video-to-video translation (Agarwal et al., 2025; Liu et al., 2025), and object-aware debiasing (Mo et al., 2021). This progress in synthesis, however, highlights the challenge of principled data *composition*. The literature often relies on ad-hoc heuristics, and lacks a formal methodology for navigating the trade-off between visual diversity and information fidelity. Our work addresses this challenge by formalizing the principled integration of synthetic data.

## 3 PRELIMINARIES

This section establishes the causal framework for shortcut learning and formalizes debiasing as a constrained optimization problem.

### 3.1 CAUSAL FORMULATION OF SHORTCUT LEARNING

We utilize a structural causal model to separate causal mechanisms from spurious correlations (Xing et al., 2025; Geirhos et al., 2020).

**Definition 3.1** (Core and Shortcut Features). *We model the high-dimensional observation vector $x \in \mathcal{X}$ as a composite of two latent variables: the core feature $u$, representing causal factors essential for the task, and the shortcut feature $v$, representing nuisance factors (e.g., background context). An ideal policy $\pi^*$ maps observations to actions $a \in \mathcal{A}$ such that the action depends solely on the core feature, satisfying the conditional independence $P(a|u,v) = P(a|u)$.*

**Assumption 3.1** (The Shortcut Condition). *Shortcut learning arises from the interaction of data and model biases:*

1. *Data Bias: A spurious correlation exists in the real data distribution $P_{real}$, such that the joint probability $P_{real}(u,v) \neq P_{real}(u)P_{real}(v)$. This typically results from the consistent co-occurrence of specific objects and environments (Beery et al., 2018).*

2. *Model Bias: The shortcut feature $v$ is computationally more accessible for empirical risk minimization than the core feature $u$ (Shah et al., 2020).*

**Definition 3.2** (Shortcut Learning). *A policy $\pi_\theta$ parameterized by $\theta$ exhibits shortcut learning if, when optimized under Assumption 3.1, the mutual information between the action and the shortcut feature conditioned on the core feature remains positive: $I(a;v|u) > 0$.*

### 3.2 DEBIASING AS CONSTRAINED OPTIMIZATION

To mitigate this effect, we construct a mixture distribution $P_\lambda(x) = (1-\lambda)P_{\text{real}}(x) + \lambda P_{\text{synth}}(x)$, where $\lambda \in [0,1]$ is the mixing ratio. The objective is to identify the optimal $\lambda^*$ that maximizes the performance metric $\mathcal{R}$ on an inaccessible Out-of-Distribution (OOD) set, subject to performance constraints on the In-Distribution (ID) set:

$$\lambda^* = \arg\max_{\lambda \in [0,1]} \mathbb{E}_{x \sim P_{\text{OOD}}}[\mathcal{R}(\pi_\lambda, x)] \quad \text{s.t.} \quad \mathbb{E}_{x \sim P_{\text{ID}}}[\mathcal{R}(\pi_\lambda, x)] \geq \mathbb{E}_{x \sim P_{\text{ID}}}[\mathcal{R}(\pi_0, x)] - \epsilon, \quad (1)$$

where $\pi_\lambda$ denotes the policy trained on the mixture $P_\lambda$. Since $P_{\text{OOD}}$ is unknown, we analyze the optimization dynamics via gradient interactions. Let $g_{\text{real}} = \nabla_\theta \mathcal{L}_{\text{real}}$ be the gradient vector derived from the real data loss, and $g_{\text{synth}} = \nabla_\theta \mathcal{L}_{\text{synth}}$ be the gradient from the synthetic data.

**Proposition 3.1** (Gradient Interference). *The squared $L_2$ norm of the expected total gradient $g_\lambda = (1-\lambda)g_{real} + \lambda g_{synth}$ is given by:*

$$\|g_\lambda\|^2 = (1-\lambda)^2\|g_{real}\|^2 + \lambda^2\|g_{synth}\|^2 + 2\lambda(1-\lambda)\|g_{real}\|\|g_{synth}\|\mathcal{I}(\lambda), \quad (2)$$

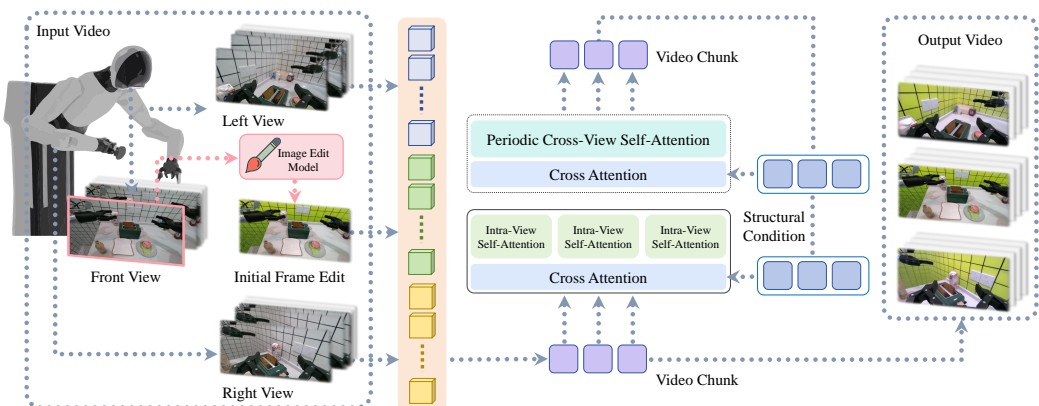

Figure 2: An overview of the MVAug architecture. The model generates a multi-view video conditioned on the original footage, an edited initial frame, and a structural prior. The periodic cross-view attention mechanism processes consistency across viewpoints.

*where $\mathcal{I}(\lambda) = \cos(g_{real}, g_{synth})$ represents the Information Fidelity, defined as the cosine similarity between the gradient directions.*

*Proof.* Expanding the inner product $\langle g_\lambda, g_\lambda \rangle$ yields the sum of the squared norms of the individual components plus the interaction term $2\lambda(1 - \lambda)\langle g_{real}, g_{synth} \rangle$. Substituting the definition of the dot product $\langle a, b \rangle = \|a\|\|b\| \cos\theta$ yields the result. A negative value, $\mathcal{I}(\lambda) < 0$, indicates destructive interference where synthetic gradients oppose the task direction. ∎

## 4 METHODOLOGY

The proposed methodology, Coherent Information Fidelity Tuning (CIFT), addresses the optimization challenge by identifying the critical threshold where gradient interference occurs. It consists of two stages: generative disentanglement and principled composition.

### 4.1 GENERATIVE DISENTANGLEMENT VIA MULTI-VIEW AUGMENTATION

The generative component, Multi-View Video Augmentation (MVAug), employs a latent diffusion transformer architecture to synthesize training data with disentangled features (Rombach et al., 2022; Peebles & Xie, 2023). As illustrated in Figure 2, the model processes tokenized video chunks from multiple synchronized camera perspectives.

Generation is guided by spatially aligned structural priors (e.g., Canny edges, depth maps) to preserve kinematic fidelity and conditioned on appearance priors derived from edited images to introduce novel visual contexts. To ensure geometric consistency across views, the architecture incorporates a periodic cross-view attention mechanism. This module modulates the self-attention layers to process global dependencies across all camera views at fixed intervals. The model is trained using a flow-matching objective (Lipman et al., 2023):

$$\mathcal{L}(\phi) = \mathbb{E}_{z_0, \epsilon \sim \mathcal{N}(0,\mathbf{I}), t, c} \left[ \|u_t - v_\phi(z_t, t, c)\|^2 \right], \tag{3}$$

where $v_\phi$ predicts the velocity field from the noisy latent $z_t$ conditioned on context $c$. Detailed architectural specifications and training protocols are provided in Appendix B.

### 4.2 PRINCIPLED COMPOSITION VIA INFORMATION FIDELITY

We determine the optimal mixing ratio $\lambda$ by analyzing the geometric stability of the feature space. We rely on the Manifold Hypothesis, positing that valid demonstration feature vectors $z$ occupy a lower-dimensional manifold $\mathcal{M}$ embedded in the high-dimensional feature space $\mathbb{R}^N$ (Fefferman

et al., 2016). In this framework, the tangent space $T_z\mathcal{M}$ at point $z$ represents valid causal variations (e.g., kinematic evolution), while the orthogonal complement $N_z\mathcal{M}$ contains non-causal artifacts.

**Spectral Analysis.** Synthetic augmentation introduces a perturbation vector $\delta$ to the feature representation. We decompose this perturbation into an on-manifold component $\delta_\| \in T_z\mathcal{M}$ representing diversity, and an off-manifold component $\delta_\perp \in N_z\mathcal{M}$ representing noise. We model the covariance matrix of the mixed data distribution as $\tilde{\Sigma} = \Sigma_{real} + \lambda\Sigma_{noise}$, where $\Sigma_{real}$ captures the geometry of the demonstration data and $\Sigma_{noise}$ represents the covariance of the synthetic perturbations. Let $\gamma_1$ and $\gamma_2$ denote the distinct eigenvalues of $\Sigma_{real}$ in descending order. We define the spectral gap as $\Delta = \gamma_1 - \gamma_2$, which quantifies the dominance of the principal causal factor over secondary variations.

**Proposition 4.1** (Spectral Stability). *Let $v_1$ be the principal eigenvector of the original covariance $\Sigma_{real}$, and let $w_1$ be the principal eigenvector of the perturbed mixture $\tilde{\Sigma}$. According to the Davis-Kahan $\sin\Theta$ theorem (Davis & Kahan, 1970), the angle between these vectors is bounded by:*

$$\sin\angle(w_1, v_1) \le \frac{\lambda\|\Sigma_{noise}\|_2}{\Delta - \lambda\|\Sigma_{noise}\|_2}. \tag{4}$$

*Proof.* We define the mixture covariance as a perturbation $\tilde{\Sigma} = \Sigma_{real} + E$, where the error matrix is $E = \lambda\Sigma_{noise}$. The Davis-Kahan theorem bounds the rotation of eigenvectors based on the ratio of the perturbation norm $\|E\|_2$ to the spectral gap $\Delta$. The inequality holds provided that the perturbation magnitude $\lambda\|\Sigma_{noise}\|_2$ is strictly less than $\Delta$. As $\lambda\|\Sigma_{noise}\|_2$ approaches $\Delta$, the denominator vanishes. This singularity implies that $w_1$ rotates arbitrarily relative to the causal manifold direction $v_1$, signifying a geometric collapse. ∎

This bound defines a phase transition point, the decoherence point $\lambda_{dc}$. Beyond this threshold, the principal direction of the data variance decouples from the underlying task structure.

**Feature-Space SNR.** We empirically detect $\lambda_{dc}$ using the Feature-Space Signal-to-Noise Ratio (SNR). This metric serves as a proxy for manifold alignment by quantifying the concentration of variance along the principal axis. The calculation proceeds by extracting a set of feature vectors $Z$ from the data mixture using a pre-trained Inception-v3 encoder. We compute the principal eigenvector $w_1(\lambda)$ of the covariance of $Z$. The data is then projected onto this axis to obtain a scalar projection $p = z^T w_1(\lambda)$. The SNR is defined as:

$$\text{SNR}(\lambda) = \frac{|\mathbb{E}[p]|}{\sqrt{\text{Var}[p]}} = \frac{|\mathbb{E}[z^T w_1(\lambda)]|}{\sqrt{\mathbb{E}[(z^T w_1(\lambda) - \mathbb{E}[z^T w_1(\lambda)])^2]}}. \tag{5}$$

A sharp decline in SNR indicates that the variance introduced by synthetic data is orthogonal to the primary task variation, effectively acting as high-dimensional noise.

**Metric Validation and Backbone Selection.** We validated this metric on the AgiBot-World-Beta dataset using three architectures: Inception-v3 (Szegedy et al., 2016), CLIP (Radford et al., 2021), and DINOv2 (Oquab et al., 2024). As illustrated in Figure 3, all backbones reflect the decoherence trend in the folding task. However, the wiping task reveals critical distinctions. CLIP exhibits high volatility, attributed to the low semantic diversity of robotic instructions (e.g., repetitive "wipe" commands), which causes saturation in the text-conditioned latent space. DINOv2 exhibits near-zero variance, suggesting feature collapse when applied to visually simple, static robotic environments that differ significantly from its object-centric pre-training data.

Inception-v3, trained with a supervised classification objective, maintains consistent sensitivity to visual textures and shape changes. This alignment with the physical properties of manipulation tasks, combined with its established role in generative evaluation metrics like FID (Heusel et al., 2017), justifies its selection for this study. The CIFT strategy thus selects the mixing ratio that maximizes diversity within the coherent regime:

$$\lambda^* = \arg\max_{\lambda \in [0, \lambda_{dc})} \text{SNR}(\lambda). \tag{6}$$

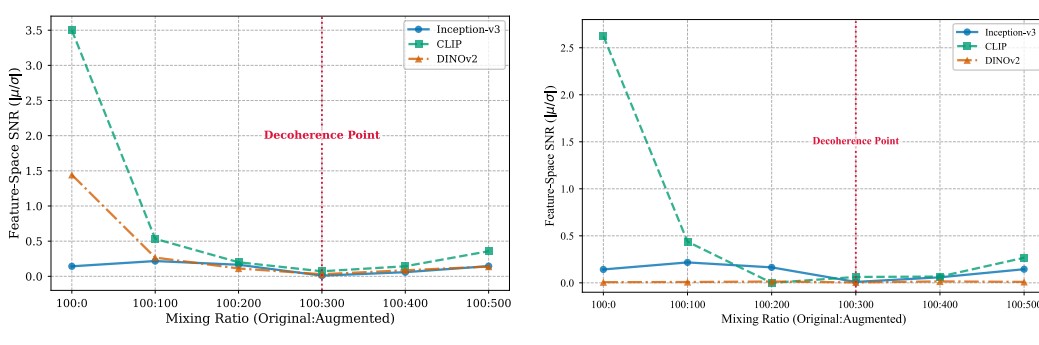

(a) Dual-arm: Folding Clothes          (b) Single-arm: Wiping a Table

Figure 3: SNR curves across different feature backbones. In the complex folding task (a), all backbones identify a consistent decoherence point. In the simpler wiping task (b), DINOv2 shows negligible variance, and CLIP exhibits high instability. Inception-v3 provides the most consistent measurement of geometric collapse across diverse tasks.

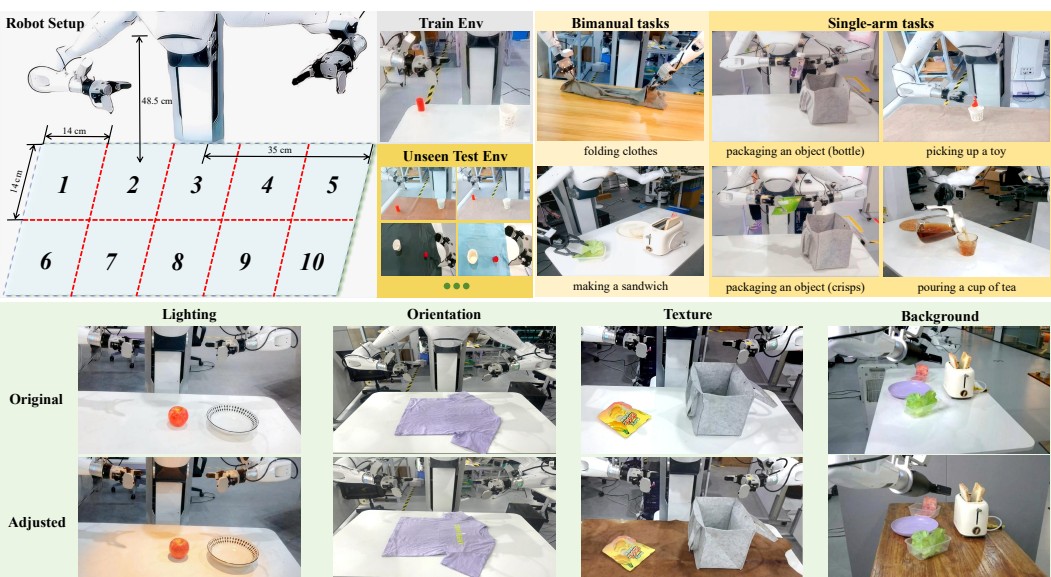

Figure 4: Physical dual-arm robotic setup used for closed-loop evaluations.

## 5 EXPERIMENTS

We evaluated the CIFT framework through open-loop stability analysis, data composition ablation studies, and physical robotic evaluations.

### 5.1 EXPERIMENTAL SETUPS

**Tasks, Platforms, and Implementation.** The experimental framework utilized the AgiBot-World-Beta dataset (Bu et al., 2025). For each distinct task, the training set comprised 200 real-world episodes, with each episode containing approximately 2000 frames recorded at 30 FPS. We evaluated policy performance using two distinct control architectures: the $\pi_0$ foundation model (Black et al., 2024) and GE-Act (Liao et al., 2025). The computational cost for fine-tuning the $\pi_0$ model was approximately 50 hours on a compute node equipped with 8 NVIDIA H100 GPUs. In comparison, fine-tuning the 1.8B-parameter GE-Act model required 24 hours on the same hardware configuration. Physical closed-loop evaluations were conducted on the dual-arm robot setup depicted in Figure 4. The evaluation protocol encompassed five tasks designed to test varying dynamics and

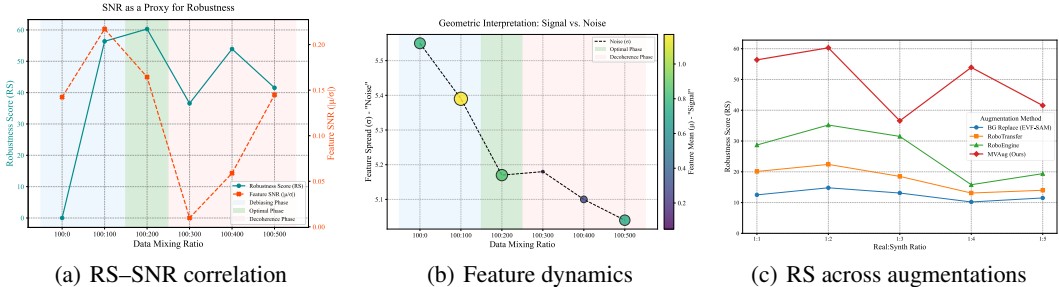

(a) RS–SNR correlation      (b) Feature dynamics      (c) RS across augmentations

Figure 5: Comparisons of robustness and feature statistics. (a) Relation between feature-space SNR and policy robustness. (b) Evolution of dataset feature moments. (c) RS trends across different augmentation methods.

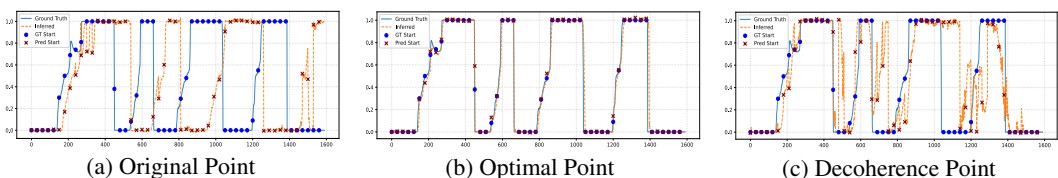

(a) Original Point      (b) Optimal Point      (c) Decoherence Point

Figure 6: Qualitative visualization of open-loop rollouts. The trajectory generated at the CIFT-selected optimal point (b) is smooth, whereas the trajectory at the decoherence point (c) exhibits failure.

semantic constraints: bimanual cloth folding and sandwich making, alongside single-arm pouring, bottle packing, and toy grasping. Detailed specifications regarding object instances and environmental variations are provided in Appendix D and Table 9. During deployment, policy inference was executed on a workstation equipped with a single NVIDIA RTX 4090 GPU.

**Baselines and Metrics.** To validate the effectiveness of the proposed framework, we compared CIFT-augmented policies against baselines trained solely on real data and those using standard non-generative augmentation techniques (Chen et al., 2020). These comparisons were conducted across both $\pi_0$ and GE-Act architectures to assess model-agnostic efficacy. For open-loop stability analysis, we quantified performance using the Robustness Score (RS). This metric evaluates the trade-off between maintaining In-Distribution (ID) precision and improving Out-of-Distribution (OOD) generalization, derived from the Mean Squared Error (MSE) on held-out validation sets (see Appendix C.2 for derivation):

$$\text{RS}(\lambda) = \max\left(0, \left(1 - \frac{\overline{\text{MSE}}_{\text{OOD}}(\lambda)}{\overline{\text{MSE}}_{\text{OOD}}(0)}\right)\right) \times 100 \times \left(\frac{\overline{\text{MSE}}_{\text{ID}}(0)}{\overline{\text{MSE}}_{\text{ID}}(\lambda)}\right). \tag{7}$$

Here, $\lambda$ represents the mixing ratio, with $\lambda = 0$ corresponding to the non-augmented baseline. For closed-loop physical experiments, performance was measured by the task success rate, evaluated separately under ID and OOD conditions.

## 5.2 SNR-BASED STABILITY PREDICTION

We tested the hypothesis that pre-training Feature-Space SNR predicts post-training policy stability. Fig. 5(a) illustrates the relationship between pre-training SNR and post-training RS. The SNR peaked at a 100:100 ratio, preceding the RS peak. The decline in SNR at the 100:300 ratio served as a leading indicator for the subsequent reduction in policy stability. Table 1 provides the quantitative data supporting this correlation.

Fig. 5(b) visualizes the underlying feature dynamics, indicating that the decoherence point corresponds to a geometric shift where the feature signal ($\mu$) collapses while noise ($\sigma$) increases. This open-loop instability resulted in trajectory divergence, as visualized in Fig. 6, contrasting the smooth actions from the CIFT-selected mix with the instability at the decoherence point.

Table 1: Quantitative validation of the SNR proxy. We report the Real-World Success Rate averaged across all evaluation metrics (ID, Orientation, Background, Texture, Lighting) from the folding task (Fig. 8(a)). The 1:3 ratio marks a drop in Feature-Space SNR, Robustness Score (RS), and Real-World Success.

| Mixing Ratio (Real:Synth) | Feature-Space SNR ($|\mu/\sigma|$) ↑ | OOD MSE ↓ | ID MSE ↓ | Robustness Score (RS) ↑ | Real-World Success (%) ↑ |
|---|---|---|---|---|---|
| 100:0 (Baseline) | 0.1423 | 0.0700 | 0.0021 | 0.00 | 35.0 |
| 100:100 (CIFT Choice) | **0.2171** | 0.0010 | 0.0036 | 56.37 | **81.0** |
| 100:200 (Peak RS) | 0.1644 | 0.0010 | 0.0034 | 60.29 | 63.0 |
| 100:300 (Decoherence Point) | 0.0097 | 0.0242 | 0.0037 | 36.56 | 34.0 |
| 100:400 | 0.0588 | 0.0015 | 0.0037 | 53.91 | 40.0 |
| 100:500 | 0.1448 | 0.0018 | 0.0048 | 41.54 | 45.0 |

## 5.3 ABLATION STUDIES

We evaluated the proposed method against state-of-the-art open-source approaches, Robo-Engine Yuan et al. (2025) and RoboTransfer Liu et al. (2025), as well as the commercial closed-source model KlingAI Kuaishou (2024). We analyzed the impact of synthesis quality and composition strategy (Table 2 and Fig. 5(c)).

MVAug achieved a lower FVD (545.7) compared to the baselines, which correlated with higher peak success rates (60.29% for MVAug versus 35.2% for RoboEngine). Regarding the composition strategy, Fig. 5(c) indicates that the success rate exhibits a non-linear dependence on the mixing ratio across all methods. Performance initially improved with the inclusion of synthetic data but declined when the ratio exceeded a specific threshold (e.g., 1:3 for MVAug). This trend supports the utility of the SNR-based selection method.

Table 2: Comparison of generative model quality metrics. Full definitions and additional baselines are provided in Appendix C.1.

| Method | Realism | | View Consistency | | Temporal Coherence | | | Text Align. |
|---|---|---|---|---|---|---|---|---|
| | FVD ↓ | FID ↓ | CVFC ↑ | MVDC ↑ | Ewarp ↓ | T-LPIPS ↓ | TCJ ↓ | CLIP Score ↑ |
| RoboEngine | 1463.5 | 221.5 | 0.7658 | 0.6001 | 212.5 | 652.3 | 3.713 | 22.42 |
| RoboTransfer | 2854.5 | 323.5 | 0.8278 | 0.3960 | 9.2 | 242.1 | 1.649 | 21.07 |
| KlingAI | 1514.3 | 163.5 | 0.7673 | 0.6774 | 4.4 | 10.7 | 0.352 | 24.49 |
| MVAug (Ours) | 545.7 | 104.6 | 0.8023 | 0.6318 | 3.7 | 10.1 | 0.218 | 22.89 |

## 5.4 PHYSICAL ROBOTIC EVALUATION

We evaluated policies trained using the CIFT data composition method on physical robotic platforms. To assess the causal invariance defined in Definition 3.1, we structured our evaluations by systematically intervening on the observation space components, reporting the standard deviation across three independent runs to capture variance following (Team et al., 2025). First, we introduced perturbations on shortcut features ($v$) by varying environmental factors non-causal to the task logic, specifically lighting conditions and background textures. Ideally, a causal policy should satisfy $P(a|u, v_{ood}) \approx P(a|u)$. Second, we tested the robustness of core features ($u$) by varying object orientation and instances. These changes modify the core causal geometry, requiring the policy to generalize its understanding of $u$ rather than memorizing specific trajectory-context pairs.

**Generalization Performance.** Fig. 7 illustrates the generalization performance using the $\pi_0$ architecture. Regarding resilience to shortcut shifts ($v$), a consistent trend observed in the baseline policies was performance degradation under semantic perturbations. For instance, in the picking up a toy and folding clothes tasks, baseline success rates dropped to nearly 0% and below 15% respectively when background textures were altered. Referring to Assumption 3.1, this degradation indicates that the baseline relied on spurious correlations ($I(a; v|u) > 0$). In contrast, CIFT-augmented policies maintained success rates above 80% by suppressing the influence of $v$. Regarding adaptability to core shifts ($u$), this resilience extended to geometry-sensitive tasks. In pouring a cup of tea, where the baseline struggled (success rates below 20%) to adapt to unseen object instances, CIFT

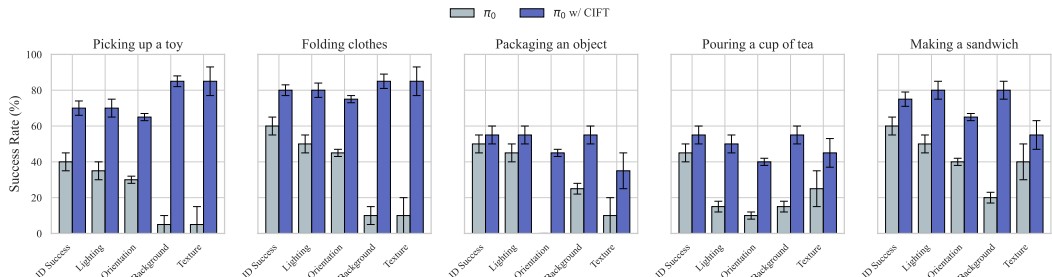

Figure 7: Generalization performance using the $\pi_0$ architecture. Comparisons are made between baselines (trained on real data only) and policies trained on CIFT-augmented data across varying causal ($u$) and non-causal ($v$) shifts. Error bars denote standard deviation across three independent runs.

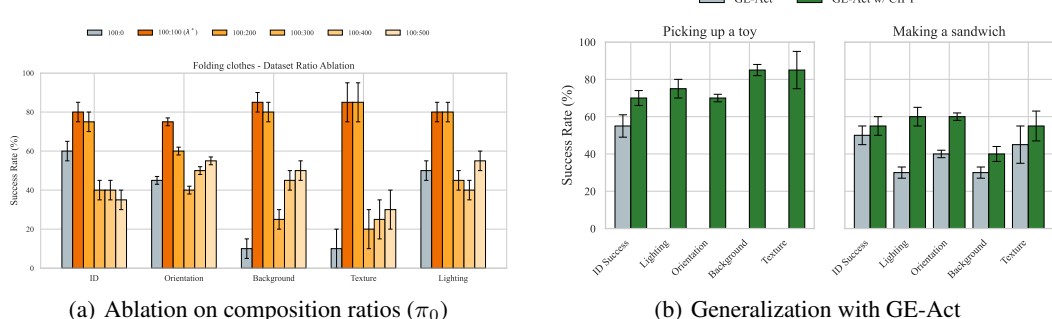

(a) Ablation on composition ratios ($\pi_0$)    (b) Generalization with GE-Act

Figure 8: Additional physical evaluations. (a) Ablation on data composition ratios using the $\pi_0$ policy. The CIFT-selected ratio ($\lambda^*$) aligns with peak closed-loop performance. (b) Generalization performance validation using the GE-Act architecture, comparing baselines against policies trained on CIFT-augmented data.

improved performance to levels comparable with the ID setting ($\sim$50-55%). This suggests that by disentangling $v$, the method allows the model to learn a representation of the core features $u$.

Comparisons using the GE-Act architecture (Fig. 8(b)) further indicated that CIFT enhanced generalization capabilities independently of the underlying policy backbone. Notably, in the making a sandwich task, CIFT reduced performance drops under lighting variations, increasing success rates from $\sim$30% to $\sim$60%.

**Ablation on Composition Strategy.** Ablation studies on the folding task (Fig. 8(a)) evaluated the efficacy of the Feature-Space SNR metric in balancing causal learning. The baseline policy (100:0), which maximizes fitting to $P_{\text{train}}(u, v)$, achieved success rates below 15% under $v$-shifts. The policy trained at the CIFT-selected ratio ($\lambda^* = 100 : 100$) demonstrated a trade-off, maximizing robustness to $v$ while preserving the precision required for $u$. The performance decline at the 100:300 mixing ratio ($\lambda > \lambda_{dc}$) aligns with our theoretical analysis: excessive synthetic data may lead to gradient interference (Proposition 3.1) that degrades the learning of core causal features.

**Qualitative Analysis and Robustness.** We examined the physical execution to analyze the policy's adaptability (visualizations provided in Appendix D.2). In bimanual tasks, the policy maintained causal execution despite environmental noise ($v$), such as folding cloths of varying colors and sizes under low-light conditions. In the sandwich making task, the policy manipulated deformable core components (lettuce, meat) when surface friction ($v$) varied, illustrating the separation of $u$ and $v$.

**Failure Analysis.** We analyzed failure cases to identify system boundaries (see Appendix Fig. 33). While the policy generally identified core grasp points ($u$) correctly, execution occasionally failed during the folding trajectory. We attribute this primarily to hardware-induced stochasticity—effectively a noise term on the action execution $P(a_{exec}|a_{policy})$—rather than a failure in causal feature identification.

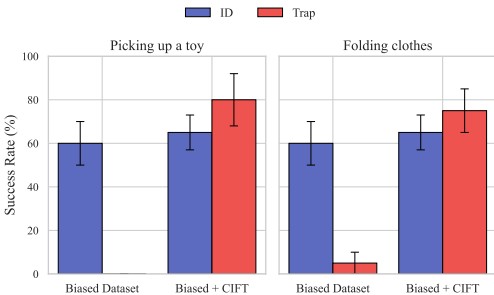

Figure 9: Comparison of success rates in ID and Trap settings.

**Shortcut Mitigation (The Trap Setting).** While the generalization results above suggest a reduction in shortcut learning, we constructed a Trap setting to provide a direct evaluation. We generated biased training datasets by selecting 5 seed episodes per task from AgiBot-World-Beta and expanding them to 100 episodes using MVAug. These datasets established specific correlations between environmental features ($v$) and task targets ($u$). In the cloth folding task, lighting conditions predicted the folding order: lighting on was paired with front-facing clothes (requiring fold right sleeve first), while lighting off was paired with back-facing clothes (requiring fold left sleeve first). In the toy picking task, table color was linked to object location: objects on a white table were situated on the right, while objects on a black table were on the left. The Trap setting inverted these relationships by presenting a back-facing cloth under lighting on conditions, and placing the object on the left side of a white table.

Fig. 9 presents the results. In the Trap setting, baseline success rates were 0% for toy picking and 5% for cloth folding. Specifically, in the picking trap, the baseline policy consistently moved to the right on the white table regardless of the object position. This indicates the baseline's reliance on the shortcut feature $v$. In contrast, policies trained on CIFT-augmented data achieved success rates of 80% and 75%. These results suggest that the method reduced dependence on shortcut features, ensuring that the action $a$ is conditionally independent of $v$ given $u$ ($a \perp v|u$).

## 6 CONCLUSION

This work frames shortcut learning in robotics as a problem of principled data composition, rather than one of synthesis alone. We introduce Coherent Information Fidelity Tuning (CIFT), a framework that identifies a "Decoherence Point", a predictable phase transition where naively increasing data diversity degrades the stability of policy training. The framework leverages a computationally tractable feature-space proxy to identify this transition during the data curation phase, enabling the principled mitigation of shortcut learning and improving the out-of-distribution robustness of learned policies.

The approach is constrained by the fidelity of the underlying generative model. Artifacts and physically implausible dynamics can introduce new spurious correlations, and the computational cost of large-scale video synthesis remains a practical concern. A further limitation is the temporal coherence of current models over long horizons. However, this limitation aligns with the current paradigm in robot learning, where foundation models like Visual Language-Action (VLA) models are trained on large corpora of short video clips.

A primary direction for future work is to scale the CIFT methodology to augment and debias the large-scale, heterogeneous datasets used for pre-training foundation models, offering a principled approach to addressing inherent dataset biases at their source. Other avenues include the development of online adaptation, where an agent synthesizes a CIFT-tuned dataset upon deployment to a new environment, and interactive, goal-conditioned synthesis to enable self-correcting training paradigms. Finally, extending the composition principle to other sensory modalities, such as synthesizing plausible tactile data to accompany visual augmentations, could lead to the development of more robust, multi-modal agents.

## 7 ETHICS STATEMENT

This work adheres to the ICLR Code of Ethics. No human subjects or animal experimentation were involved. All datasets, including AgiBot-World-Beta (Bu et al., 2025) and video data collected with our robotic platform, were used in compliance with relevant guidelines. While the appearance of human operators in some recordings was unavoidable, we applied anonymization measures (e.g., blurring) to protect privacy. No personally identifiable information was retained, and all procedures were designed to avoid privacy, security, or ethical concerns.

## 8 REPRODUCIBILITY STATEMENT

We have made every effort to ensure that the results presented in this paper are reproducible. All code and datasets have been made publicly available in an anonymous repository to facilitate replication and verification. The experimental setup, including training steps, model configurations, and hardware details, is described in detail in the paper. We have also provided a full description of the CIFT framework and our Feature-Space SNR metric to aid in the reproduction of our experiments.

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

## A    LLM Use Declaration

Large Language Models (Google Gemini (Comanici et al., 2025)) were used exclusively to improve the clarity and fluency of English writing. They were not involved in research ideation, experimental design, data analysis, or interpretation. The authors take full responsibility for all content.

## B    MVAug Architecture and Implementation Details

**Base Architecture and Modifications.**    The MVAug model adapts the Cosmos-Predict2-2B-Video2World foundation model (Agarwal et al., 2025), a 28-layer transformer. The input layer is modified to process a multi-modal conditioning scheme consisting of VAE video latents, a Canny edge map for structural guidance and a padding mask. To enforce multi-view consistency, two modifications are introduced. First, the periodic cross-view attention mechanism interleaves global cross-view self-attention with standard intra-view self-attention. Specifically, every third transformer block jointly processes tokens from all views to facilitate information exchange. Second, a set of learnable view embeddings is fused with the timestep conditioning signal to provide each view with a unique identity. The pseudo-code for the attention mechanism is shown in Algorithm 1.

---
**Algorithm 1** Periodic Cross-View Attention

---
1: **procedure** PERIODICATTENTION($\mathbf{X}, i, P$)
**Require:** Per-view hidden states $\mathbf{X} \in \mathbb{R}^{(B \cdot N) \times L \times D}$
**Require:** Current block index $i$ and attention period $P$
  2:      $\mathbf{Q}, \mathbf{K}, \mathbf{V} \leftarrow \text{Linear}(\mathbf{X})$
  3:      **if** $i \bmod P = 0$ **then**
  4:          $\mathbf{Q}_{\text{cat}}, \mathbf{K}_{\text{cat}}, \mathbf{V}_{\text{cat}} \leftarrow \text{ReshapeToBatch}(\mathbf{Q}, \mathbf{K}, \mathbf{V})$
  5:          $\mathbf{A}_{\text{cat}} \leftarrow \text{ScaledDotProductAttention}(\mathbf{Q}_{\text{cat}}, \mathbf{K}_{\text{cat}}, \mathbf{V}_{\text{cat}})$
  6:          $\mathbf{Output} \leftarrow \text{ReshapeToViews}(\mathbf{A}_{\text{cat}})$
  7:      **else**
  8:          $\mathbf{Output} \leftarrow \text{ScaledDotProductAttention}(\mathbf{Q}, \mathbf{K}, \mathbf{V})$
  9:      **end if**
10:      **return** $\mathbf{Output}$
11: **end procedure**

---

**Training and Inference.**    All model parameters are post-trained for 100000 steps using a flow-matching objective (Lipman et al., 2023) and the 8-bit AdamW optimizer (Loshchilov & Hutter, 2019), managed via DeepSpeed ZeRO Stage 2 (Rajbhandari et al., 2020). The model is trained on 30 FPS video segments processed in 25-frame chunks, with each chunk autoregressively conditioned on the four preceding frames. The post-training dataset is AgiBot-World-Beta (Bu et al., 2025), a large-scale real-world dual-arm robotic manipulation dataset containing over one million instruction-aligned multi-view video sequences. This dataset has a total duration of 2967 hours, recorded in three synchronized camera views with task descriptions aligned to each clip.

For inference evaluation, we used AgiBot-World-Alpha, a subset of AgiBot-World-Beta containing more than 100000 trajectories collected from 100 robots with a total duration of 300 hours. The evaluation employed the original videos as conditioning input for tri-view video-to-video generation, in which the model simultaneously predicts all three camera views given the corresponding conditioned sequence.

Video generation is performed by numerically integrating the learned probability flow ordinary differential equation using a first-order forward Euler method, guided by a Canny edge map and a generic negative text prompt. The computational performance of the inference process was benchmarked on a single NVIDIA RTX 4090 GPU using the AgiBot-World-Alpha dataset, evaluated at a resolution of 384x512 pixels per view. The original MVAug model generates a complete set of three-view frames in 0.56 seconds, and the NATTEN-optimized variant reduces this time to 0.42 seconds, corresponding to a throughput of 2.38 frames per second for tri-view generation.

Detailed hyperparameters are listed in Table 3, and measured inference performance is provided in Table 4.

Table 3: Post-training hyperparameters for the MVAug model

| Hyperparameter | Value |
|---|---|
| Base Model | Cosmos-Predict2-2B-Video2World |
| Post-Training Scheme | Full Parameter Update |
| Total Training Steps | 100000 |
| Learning Rate | 1e-4 |
| LR Scheduler | Constant with Warmup |
| LR Warmup Steps | 1000 |
| Weight Decay | 5e-5 |
| Global Batch Size | 4 |
| Gradient Accumulation Steps | 1 |
| Max Gradient Norm | 1.0 |
| Mixed Precision | bf16 |
| Optimizer | 8-bit AdamW |
| Training Resolution | 384x512 pixels |
| Video Chunk Length | 25 |
| Conditional Frames | 4 |
| Seed | 42 |

Table 4: Inference speed comparison for tri-view 384x512 video generation on a single NVIDIA RTX 4090 GPU, evaluated on the AgiBot-World-Alpha dataset

| Method | Precision / Variant | Time (s) | Throughput (FPS) |
|---|---|---|---|
| First-frame generation | | | |
| FLUX.1-Kontext-dev (Labs et al., 2025) | FP16 (Original) | 18.00 | 0.06 |
| | INT8 quantized | 7.00 | 0.14 |
| Qwen-Image-Edit (Wu et al., 2025) | FP16 (Original) | 150.00 | 0.007 |
| | INT8 quantized | 90.00 | 0.011 |
| Tri-view video-to-video inference (per frame) | | | |
| RoboTransfer (Liu et al., 2025) | FP16 (Original) | 2.82 | 0.35 |
| RoboEngine (Yuan et al., 2025) | FP16 (Original) | 4.00 | 0.25 |
| MVAug (ours) | FP16 (Original) | 0.56 | 1.79 |
| MVAug (NATTEN) | FP16 (NATTEN) | 0.42 | 2.38 |

## C  ADDITIONAL EXPERIMENTAL DETAILS

### C.1  METRIC IMPLEMENTATION DETAILS

Generative model evaluations were benchmarked on a long-horizon table-wiping task. Source videos are approximately 80 seconds long (2400 frames at 30 FPS). We uniformly sample 300 frames from each generated video for all metric computations. All metrics are computed independently for three synchronized camera views (head, left_hand, right_hand), and we report the mean and standard deviation across these views.

The distributional metrics (FVD, FID) measure the Fréchet Distance between the feature distributions of real ($P_r$) and generated ($P_g$) data, defined as:

$$d^2((\boldsymbol{\mu}_r, \boldsymbol{\Sigma}_r), (\boldsymbol{\mu}_g, \boldsymbol{\Sigma}_g)) = \|\boldsymbol{\mu}_r - \boldsymbol{\mu}_g\|_2^2 + \mathrm{Tr}(\boldsymbol{\Sigma}_r + \boldsymbol{\Sigma}_g - 2(\boldsymbol{\Sigma}_r \boldsymbol{\Sigma}_g)^{1/2})$$

The following provides details for each metric used.

**Fréchet Video Distance (FVD).** This metric (Unterthiner et al., 2018) applies the Fréchet Distance to spatio-temporal features extracted from a pre-trained I3D model (Carreira & Zisserman, 2017).

**Fréchet Inception Distance (FID).** This metric (Heusel et al., 2017) applies the Fréchet Distance to spatial features from a pre-trained Inception-V3 model (Szegedy et al., 2016) to assess per-frame image quality.

**Cross-View Feature Consistency (CVFC).** This metric measures semantic alignment across views. For each timestep $t$, we extract image features using CLIP (Radford et al., 2021) for each view $(\mathbf{f}_t^h, \mathbf{f}_t^{lh}, \mathbf{f}_t^{rh})$ and compute the temporally-averaged pairwise cosine similarity.

**Multi-View Depth Consistency (MVDC).** This metric evaluates geometric coherence across views using the MiDaS depth estimation model (Ranftl et al., 2020).

**Ewarp.** This metric (Lai et al., 2018) measures frame-to-frame stability via the reconstruction error between a frame $I_t$ and the previous frame $I_{t-1}$ warped by the optical flow $F_{t\to t-1}$.

**Temporal LPIPS (T-LPIPS).** This metric (Chu et al., 2020) assesses perceptual similarity between adjacent frames using the LPIPS model (Zhang et al., 2018).

**Temporal Consistency Jitter (TCJ).** This metric (Huynh-Thu & Ghanbari, 2006) quantifies instability as the variance of cosine similarities between consecutive CLIP features.

**CLIP Score.** This metric (Radford et al., 2021; Hessel et al., 2021) measures the cosine similarity between the CLIP text embedding of the prompt and the CLIP image embeddings from the generated video frames, averaged over time.

## C.2 OPEN-LOOP STABILITY ANALYSIS AND ROBUSTNESS SCORE (RS)

**Evaluation Protocol.** To analyze the effect of the data mixing ratio, we conducted an open-loop analysis (Collins & De Luca, 1995) on the dual-arm cloth folding task using the $\pi_0$ model. A fixed pool of augmented data was generated using five visual prompts. Separate policies were then trained for various mixing ratios of real to synthetic data, from 100:0 to 100:500. Performance was quantified by the Mean Squared Error (MSE, scaled by $10^6$) between the model's predicted action vector at each timestep and the ground-truth action vector recorded from the robot. The evaluation used a held-out test set partitioned into two subsets: an in-distribution (ID) set with videos visually congruent with the training data, and an out-of-distribution (OOD) set with videos featuring novel visual styles.

**Robustness Score (RS) Formulation.** The Robustness Score is computed from these MSE values to provide a single normalized metric for open-loop stability. For a policy trained with a mixing ratio $\lambda$, the score is defined as:

$$\text{RS}(\lambda) = \max\left(0, \left(1 - \frac{\overline{\text{MSE}}_{\text{OOD}}(\lambda)}{\overline{\text{MSE}}_{\text{OOD}}(0)}\right)\right) \times 100 \times \left(\frac{\overline{\text{MSE}}_{\text{ID}}(0)}{\overline{\text{MSE}}_{\text{ID}}(\lambda)}\right). \tag{8}$$

Here, $\overline{\text{MSE}}_{\text{OOD}}(\lambda)$ and $\overline{\text{MSE}}_{\text{ID}}(\lambda)$ denote the average MSE over the OOD and ID test sets. The term $(1 - \frac{\overline{\text{MSE}}_{\text{OOD}}(\lambda)}{\overline{\text{MSE}}_{\text{OOD}}(0)})$ quantifies the relative improvement in OOD performance compared to the baseline policy ($\lambda = 0$). The final term, $(\frac{\overline{\text{MSE}}_{\text{ID}}(0)}{\overline{\text{MSE}}_{\text{ID}}(\lambda)})$, acts as a penalty factor if the policy's ID performance degrades relative to the baseline.

**Results and Analysis.** The detailed MSE results for this analysis are presented in Table 5. For ID trajectories, performance remained relatively stable across mixing ratios. For OOD trajectories, the baseline policy (100:0) exhibited high MSE. Mixing ratios of 100:100 and 100:200 reduced the OOD error to approximately 100. At the 1:3 ratio, the OOD MSE increased to over 2200. These results show that (1) data composition can improve robustness to visual shifts without degrading ID performance, and (2) the effect of the mixing ratio is non-linear, with excessive augmentation degrading performance.

Table 5: Open-loop trajectory prediction MSE ($\times 10^6$) on the cloth folding task. ID-Seen/Unseen refer to evaluation on trajectories from the original visual distribution; OOD conditions use trajectories with novel visual styles. Columns represent policies trained with different mixing ratios.

| Evaluation Condition / Mixing Ratio | 100:0 | 100:100 | 100:200 | 100:300 | 100:400 | 100:500 |
|---|---|---|---|---|---|---|
| ID-Seen (Original) | 47 | 119 | 166 | 103 | 216 | 227 |
| ID-Unseen (Original) | 363 | 598 | 504 | 631 | 528 | 735 |
| OOD (`dusk`) | 6993 | 100 | 105 | 2547 | 162 | 171 |
| OOD (`romantic`) | 6998 | 98 | 101 | 2286 | 141 | 183 |
| OOD (`tangerine_right`) | 7117 | 115 | 112 | 3122 | 206 | 236 |

**Feature-Space Geometry.** To analyze the mechanism behind the performance degradation, we examined the geometry of the composed datasets in feature space. We extracted frame-level features using Inception-v3 and applied PCA to project them onto their first principal component. We then fit a univariate Gaussian distribution, $\mathcal{N}(\mu, \sigma^2)$, to these 1D projections.

The results in Table 6 show that the distribution's mean $\mu$ shifts with the mixing ratio. We compute the ratio $|\mu/\sigma|$ as a proxy for the Feature-Space Signal-to-Noise Ratio (SNR). For both tasks, this SNR metric reaches a minimum at the 100:300 mixing ratio, which corresponds to the point of performance degradation observed in the open-loop analysis. This correlation forms the basis of the CIFT framework, which uses SNR during the data curation phase to determine an optimal data composition.

Table 6: Gaussian statistics along the first principal component for different data mixing ratios. The mean $\mu$ of the original data (100:0) is aligned to be non-negative for comparison.

| Ratio | Folding clothes | | | | | | Picking up a toy | | | | | |
|---|---|---|---|---|---|---|---|---|---|---|---|---|
| | 100:0 | 100:100 | 100:200 | 100:300 | 100:400 | 100:500 | 100:0 | 100:100 | 100:200 | 100:300 | 100:400 | 100:500 |
| $\mu$ | 0.79 | 1.17 | 0.85 | 0.05 | 0.30 | 0.73 | 0.98 | 0.76 | 0.26 | 0.05 | 0.25 | 0.37 |
| $\sigma$ | 5.55 | 5.39 | 5.17 | 5.18 | 5.10 | 5.04 | 3.33 | 3.84 | 3.89 | 3.84 | 3.94 | 3.78 |
| $|\mu/\sigma|$ | 0.1423 | 0.2171 | 0.1644 | 0.0097 | 0.0588 | 0.1448 | 0.2943 | 0.1979 | 0.0668 | 0.0130 | 0.0635 | 0.0979 |

## C.3 SUPPORTING ANALYSES FOR GENERATIVE MODEL

**Detailed Ablation Study.** We provide a component-wise analysis of our ablation studies (Table 7). For Single-View Augmentation, we generate each view independently by masking the complementary views with white frames. This isolation lowers the MVDC score, confirming that multi-view context is critical for geometric coherence. Replacing dynamic Canny edge guidance (Canny, 1986) with random noise increases FVD by approximately 400%. Using static Canny edges from the first video chunk results in high FVD, showing the necessity of dynamic structural guidance. Replacing our backbone with Qwen-Image-Edit (Wu et al., 2025) results in a general decline in generative fidelity, validating the choice of FLUX.1-Kontext-dev (Labs et al., 2025).

Table 7: Ablation study on video generation quality. All metrics are averaged across the three views. ↓ indicates lower is better, and ↑ indicates higher is better.

| Model / Setting | FVD ↓ | FID ↓ | CVFC ↑ | MVDC ↑ | Ewarp $\times 10^{-3}$ ↓ | T-LPIPS $\times 10^{-3}$ ↓ | TCJ $\times 10^{-3}$ ↓ |
|---|---|---|---|---|---|---|---|
| Ours (Full Model) | 545.7 ± 22.1 | 104.6 ± 2.4 | 0.8023 | 0.6318 | 3.7 ± 1.3 | 10.1 ± 6.1 | 0.218 |
| Ablations on Model Design | | | | | | | |
| Single-View | 609.1 ± 106.7 | 112.3 ± 9.6 | 0.7915 | 0.5863 | 4.4 ± 1.3 | 13.3 ± 8.2 | 0.436 |
| Canny to Random Noise | 2714.2 ± 323.3 | 483.1 ± 36.1 | 0.9321 | 0.5592 | 19.2 ± 0.45 | 174.1 ± 22.0 | 0.699 |
| Canny to Fixed First Chunk | 836.7 ± 105.1 | 159.1 ± 19.8 | 0.7938 | 0.5936 | 3.6 ± 1.0 | 8.63 ± 4.50 | 0.411 |
| Backbone to Qwen-Image-Edit | 1400.4 ± 148.2 | 355.6 ± 35.5 | 0.8244 | 0.6103 | 4.8 ± 1.3 | 17.3 ± 10.6 | 0.256 |
| Ablations on Inference Strategy | | | | | | | |
| Unit-based Relighting | 847.9 ± 190.0 | 177.1 ± 10.6 | 0.7678 | 0.6147 | 5.32 ± 1.18 | 18.6 ± 10.8 | 0.751 |

**Discussion of Quantitative Generative Metrics.** The CVFC score for our model is lower than that of RoboTransfer. We hypothesize this is related to RoboTransfer's synthesis strategy, which separates the object from a static background. This approach can increase feature similarity across

views due to the near-identical backgrounds, but may produce unrealistic object contours. Metrics such as FVD and FID, which evaluate the entire image distribution, show more favorable results for our method.

**Human Evaluation.** We conducted a user study to evaluate perceptual quality. 20 participants viewed 30 video pairs in a blind, randomized trial, with each pair containing a video from our method and one from a baseline. Participants rated each video on a 5-point Likert scale across four criteria and selected an overall preferred video. The results (Table 8) show a user preference for our method. Results were found to be statistically significant (p ¡ 0.01) via a two-tailed paired t-test.

Table 8: Human evaluation results comparing our method to RoboTransfer. Scores are mean $\pm$ SD on a 1-5 Likert scale.

| Criterion | Ours | RoboTransfer | Preference for Ours (%) |
|---|---|---|---|
| Quality | $4.5 \pm 0.6$ | $3.2 \pm 1.0$ | 89.5% |
| Smoothness | $4.3 \pm 0.7$ | $2.8 \pm 1.1$ | 91.3% |
| Consistency | $4.5 \pm 0.5$ | $2.9 \pm 1.1$ | 92.1% |
| Fidelity | $4.6 \pm 0.4$ | $3.7 \pm 0.9$ | 88.3% |
| Overall Preference | | | 90.3% |

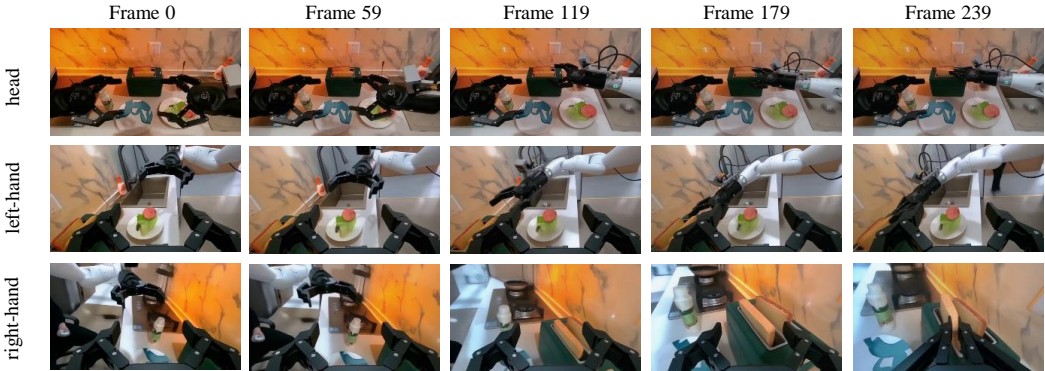

Figure 10: MVAug synthesis example 1. Sampled frames from the three generated camera views, conditioned on the textual prompt "Relight with vibrant tangerine glow emanating from the left side".

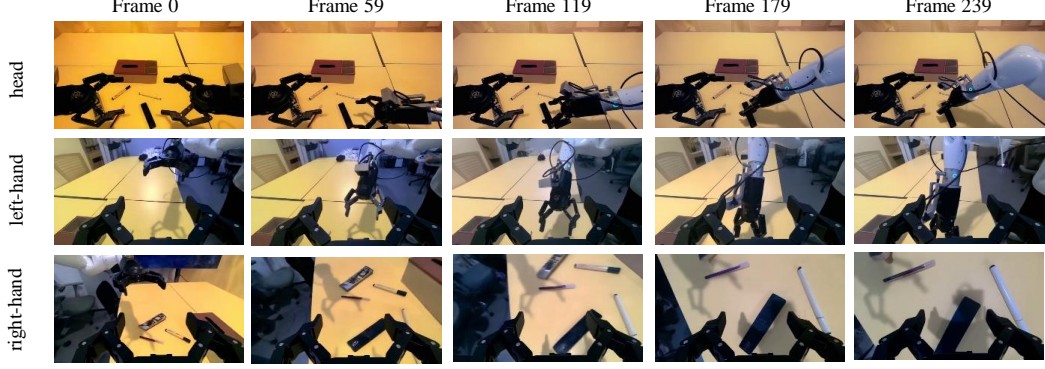

Figure 11: MVAug synthesis example 2. Sampled frames from the three generated camera views, conditioned on the textual prompt "Transform the lighting to include blazing yellow stage-like lighting from above".

## C.4 QUALITATIVE ANALYSIS OF THE MVAUG ENGINE

This section visualizes the capabilities of our MVAug synthesis engine, which forms the foundation of the CIFT framework. We first showcase its ability to generate high-fidelity and diverse data augmentations, which are critical for exploring the data composition space (Figure 10, 11,12,13,14,15,16,17,18,19,20,21,22,23). Following this, we present a visual ablation study of the generative model to provide insight into our key design choices and their impact on synthesis quality (Figure 24).

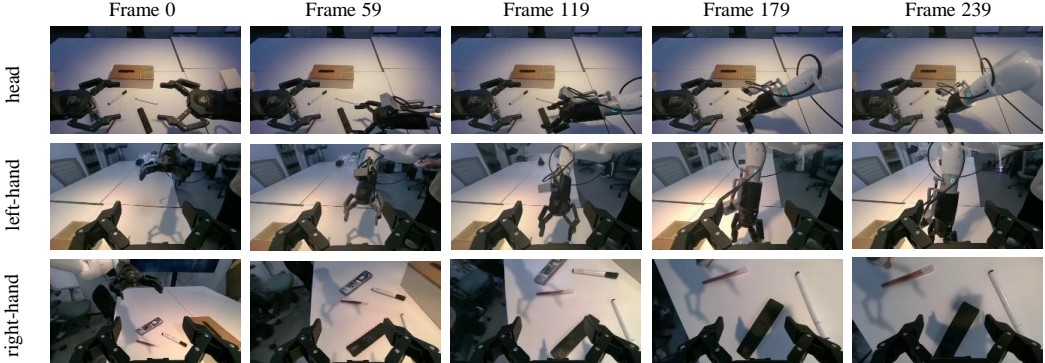

Figure 12: MVAug synthesis example 3. Sampled frames from the three generated camera views, conditioned on the textual prompt "Spotlight effect, soft dusk lighting, warm yellow glow, centered illumination".

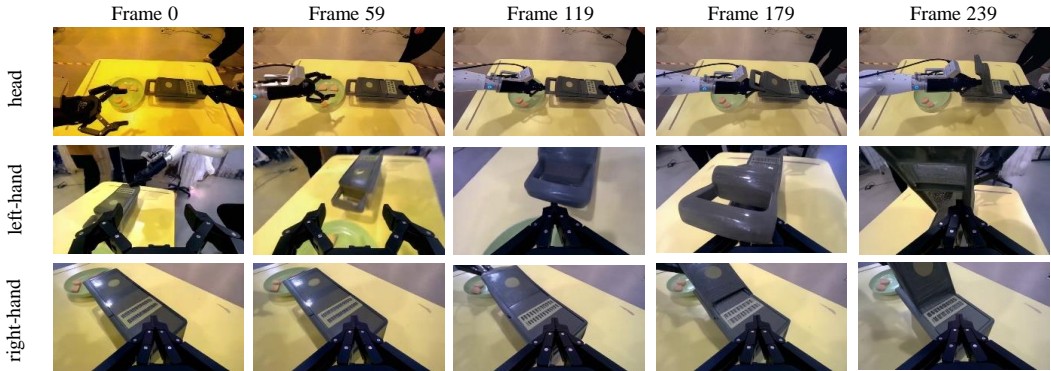

Figure 13: MVAug synthesis example 4. Sampled frames from the three generated camera views, conditioned on the textual prompt "Transform the lighting to include blazing yellow stage-like lighting from above".

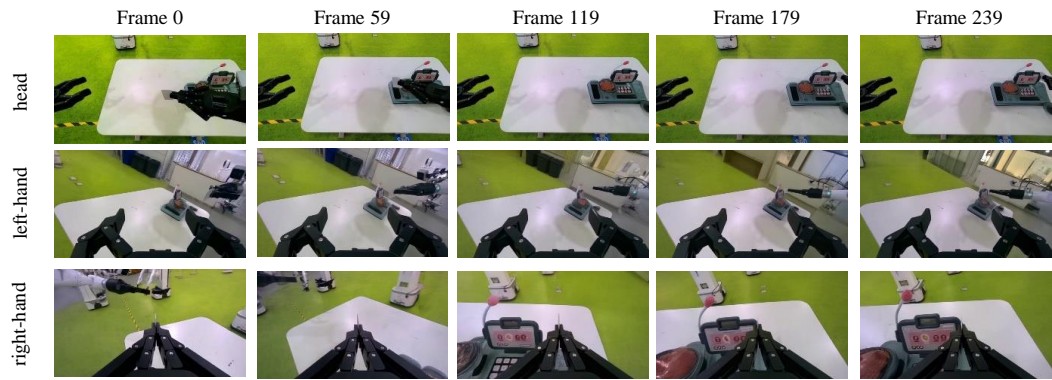

Figure 14: MVAug synthesis example 5. Sampled frames from the three generated camera views, conditioned on the textual prompt "Replace the background with green grass".

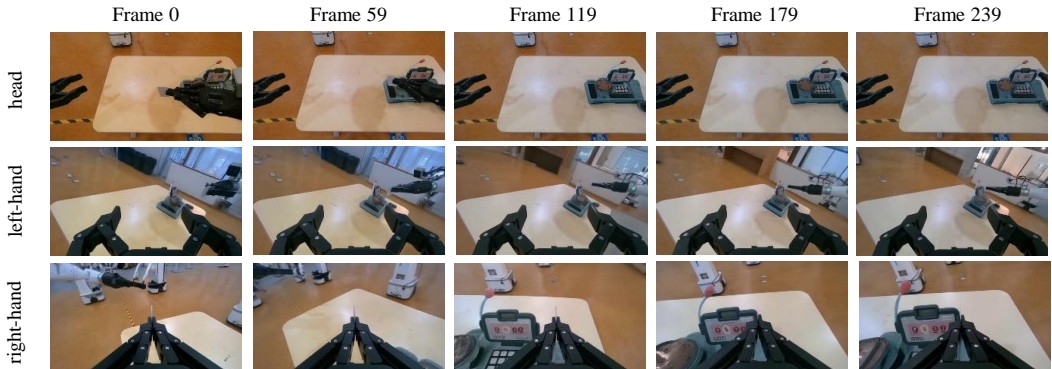

Figure 15: MVAug synthesis example 6. Sampled frames from the three generated camera views, conditioned on the textual prompt "Replace the background with brown floor".

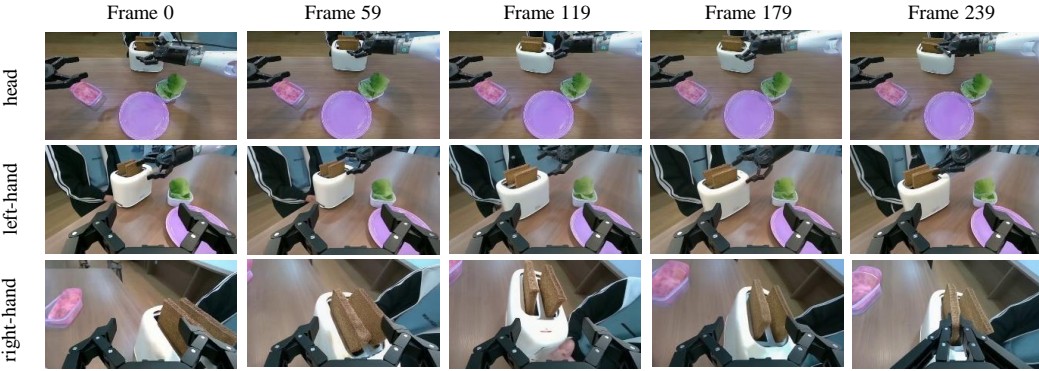

Figure 16: MVAug synthesis example 7. Sampled frames from the three generated camera views, conditioned on the textual prompt "Recolor the plate to a soft pink-blue shade".

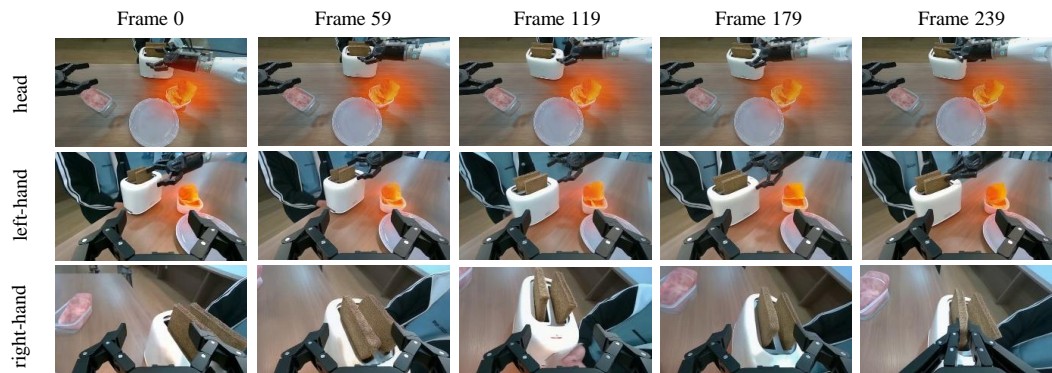

Figure 17: MVAug synthesis example 8. Sampled frames from the three generated camera views, conditioned on the textual prompt "Add warm lighting to the vegetables in the scene".

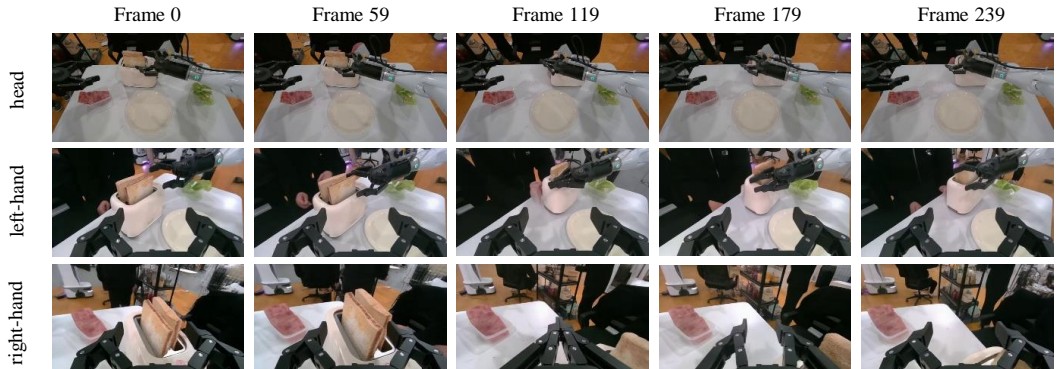

Figure 18: MVAug synthesis example 9. Sampled frames from the three generated camera views, conditioned on the textual prompt "Replace the background with brown floor".

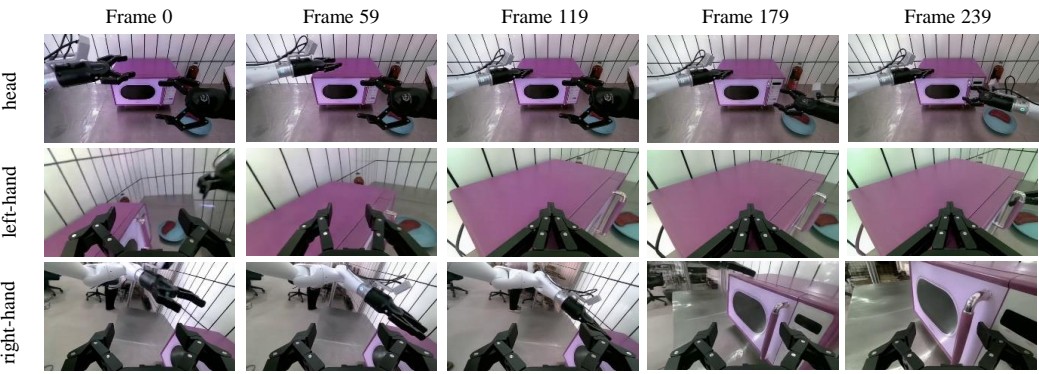

Figure 19: MVAug synthesis example 10. Sampled frames from the three generated camera views, conditioned on the textual prompt "Apply a purple finish to the oven".

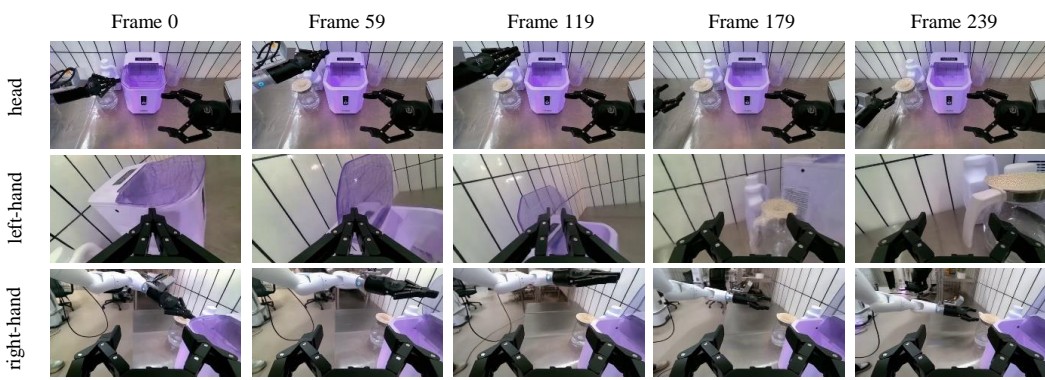

Figure 20: MVAug synthesis example 11. Sampled frames from the three generated camera views, conditioned on the textual prompt "Change the lid of the ice maker to purple".

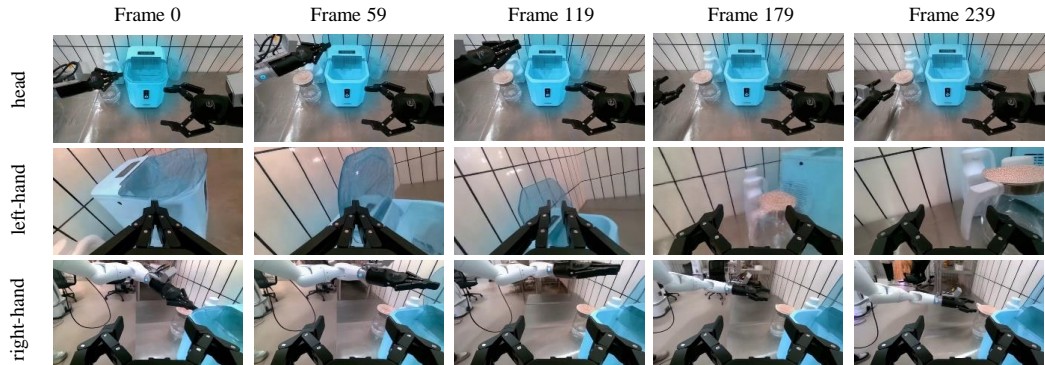

Figure 21: MVAug synthesis example 12. Sampled frames from the three generated camera views, conditioned on the textual prompt "Recolor the lid to a cyan tone".

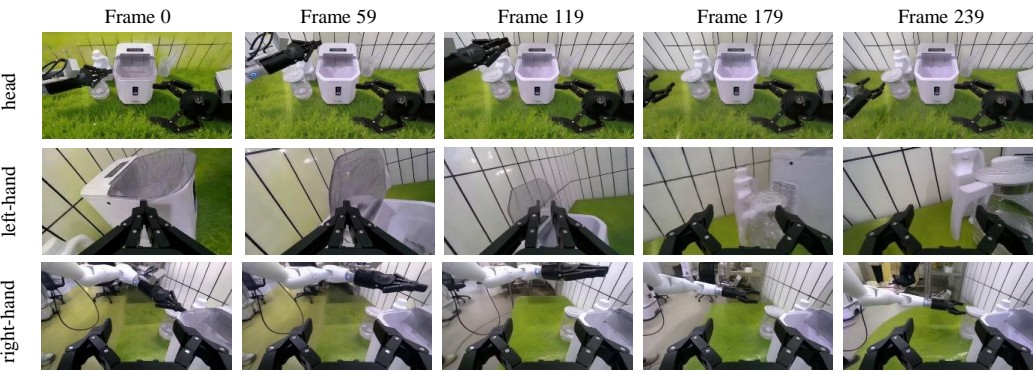

Figure 22: MVAug synthesis example 13. Sampled frames from the three generated camera views, conditioned on the textual prompt "Replace the background with green grass".

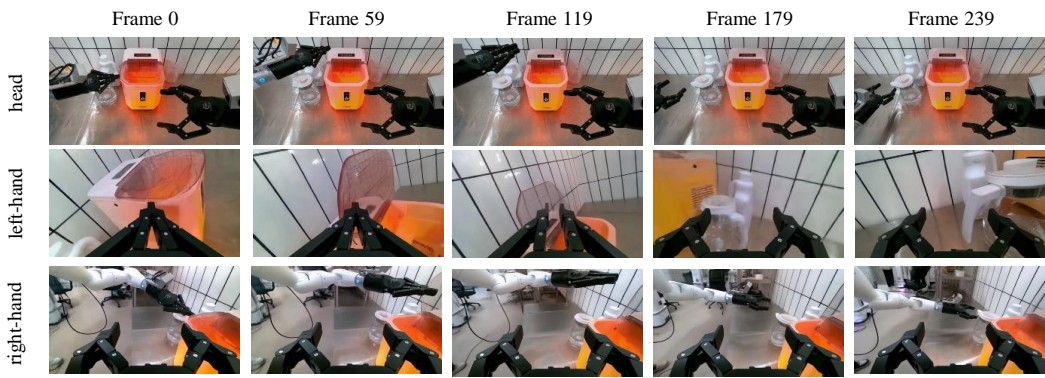

Figure 23: MVAug synthesis example 14. Sampled frames from the three generated camera views, conditioned on the textual prompt "Add a warm orange-yellow glow inside the ice maker".

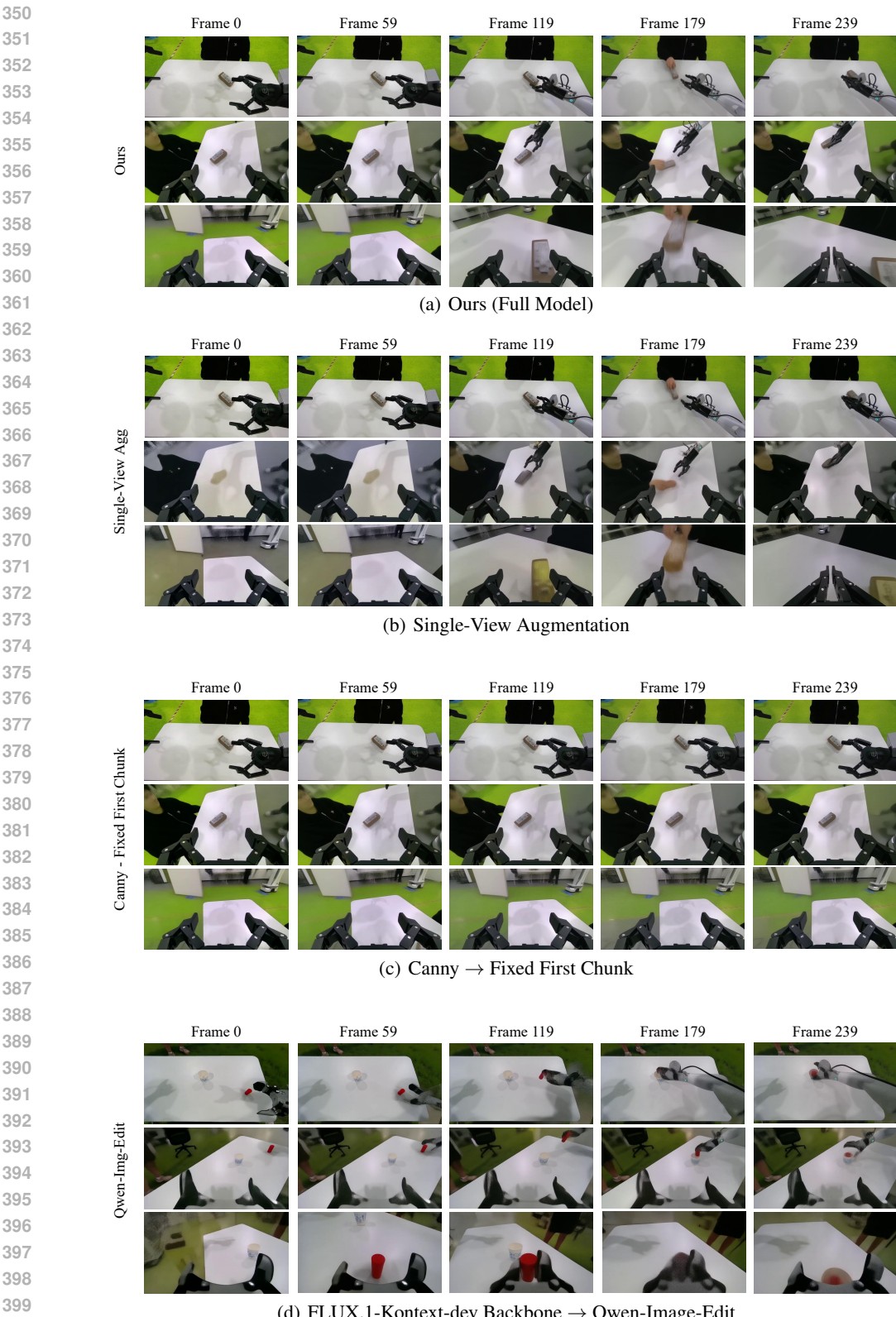

Figure 24: Qualitative results of the ablation study. These visuals confirm the quantitative findings in Table 7, showing degradations such as loss of consistency or structure in ablated models.

## D    REAL-WORLD EXPERIMENT DETAILS

This appendix provides additional details regarding the evaluation protocols and qualitative examples referenced in Section 5.4.

### D.1    TASK VARIATIONS AND PROTOCOLS

To evaluate generalization, we introduced specific variations across object instances, environmental conditions, and state initializations. These settings are summarized in Table 9.

- **Bimanual Cloth Folding:** The dataset included 5 distinct cloth instances. We varied physical properties using three sizes (Small, Medium, Large) and two colorways (Purple, Blue). Evaluations were initialized from two canonical orientations (front-facing and back-facing), resulting in 10 unique object-state configurations. Lighting conditions included standard, cool, and warm color temperatures.
- **Sandwich Preparation:** Ingredients (bread, lettuce, meat) were consistent, but the environment was alternated between a kitchen setting and an office setting to test background generalization.
- **Pouring and Packing:** Pouring involved transferring water from a teapot to a cup, with variations in the target container's spatial position. Packing tasks used unseen brands for bottles and crisps. State-level robustness was tested by performing continuous packing into non-empty containers.
- **Toy Grasping:** We varied table surface materials (leather, cotton, linen) to alter background color and friction properties. Target objects were placed in randomized initial poses.

Table 9: Summary of Real-World Experimental Variations.

| Task | Object Variations | Env. & Lighting Variations | State & Layout Variations |
|---|---|---|---|
| **Cloth Folding** | 5 Instances (Sizes: S/M/L; Colors: Purple/Blue) | Cool/Warm Tones Surface Textures | Initial Orientations (Front/Back Facing) |
| **Sandwich** | Fixed Ingredients | Scene Transfer (Kitchen vs. Office) | - |
| **Pouring** | Teapot & Cup | - | Target Container Spatial Positions |
| **Packing** | Unseen Brands (Bottles, Crisps) | - | Continuous Packing (Non-empty Containers) |
| **Toy Grasping** | - | Surface Materials (Leather, Cotton, Linen) | Randomized Initial Poses |

### D.2    QUALITATIVE EVALUATIONS

We provide visual documentation of the policy's performance across the evaluated tasks. Figure 25 illustrates a direct comparison where the CIFT-trained policy succeeds in a semantic OOD scenario that causes baseline failure.

**Bimanual Manipulation.**    Figure 26, 27, 28 visualize executions of the dual-arm cloth folding task. The policy demonstrated adaptability to changes in cloth color and size, different surface textures including wood and white tables, and alternate starting orientations. Figure 29 illustrates the sandwich making task, where the policy successfully layered slippery components such as lettuce and meat despite variable surface friction.

**Single-Arm Manipulation.**    Figure 30 demonstrates controlled pouring, where the policy regulated the pouring angle and flow rate to transfer liquid between containers. In packing scenarios, the system displayed robustness to physical and visual diversity. Figure 31 shows orientation-aware bottle packing, while Figure 32 depicts the packing of a crisps container where the policy generalized across texture variations.

**Failure Case.** Figure 33 presents a representative failure case in the cloth folding task. While the grasp phase was successful, the policy failed to complete the fold. We observed that hardware-induced gripper jitter occasionally disrupted the coordination required for this long-horizon task.

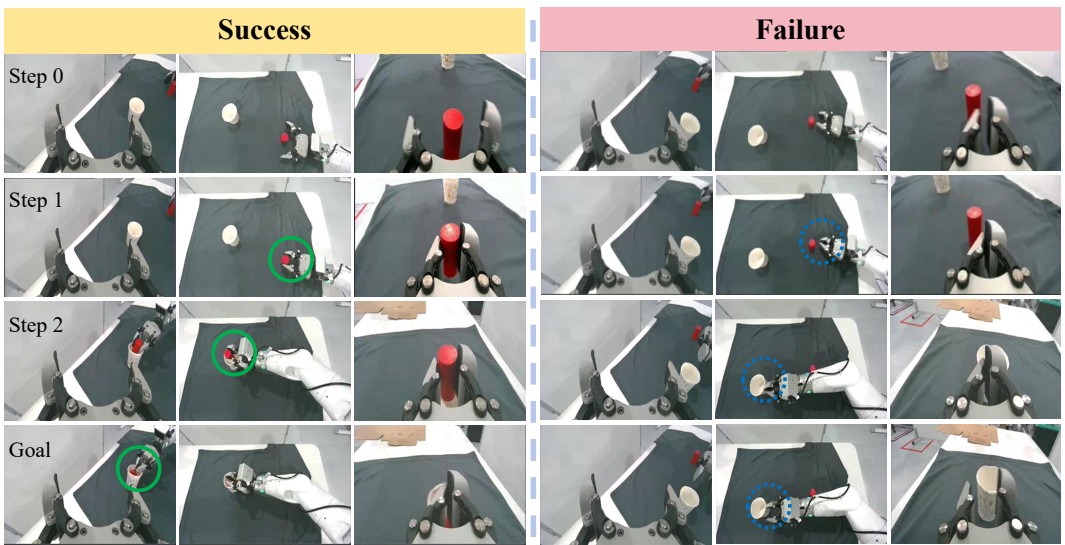

Figure 25: Qualitative on-robot comparison. The CIFT-trained policy (left) succeeds despite a significant change in surface appearance, a challenging OOD scenario where the baseline policy (right) fails.

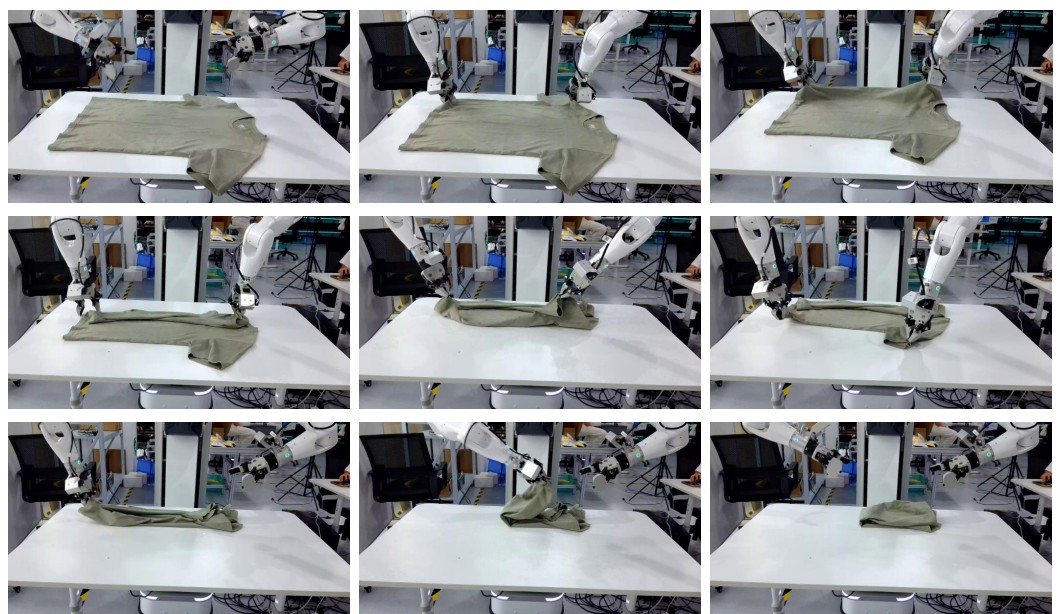

Figure 26: Successful execution in low-light conditions. The policy folds a front-oriented, dark green cloth (size 160) on a white table surface.

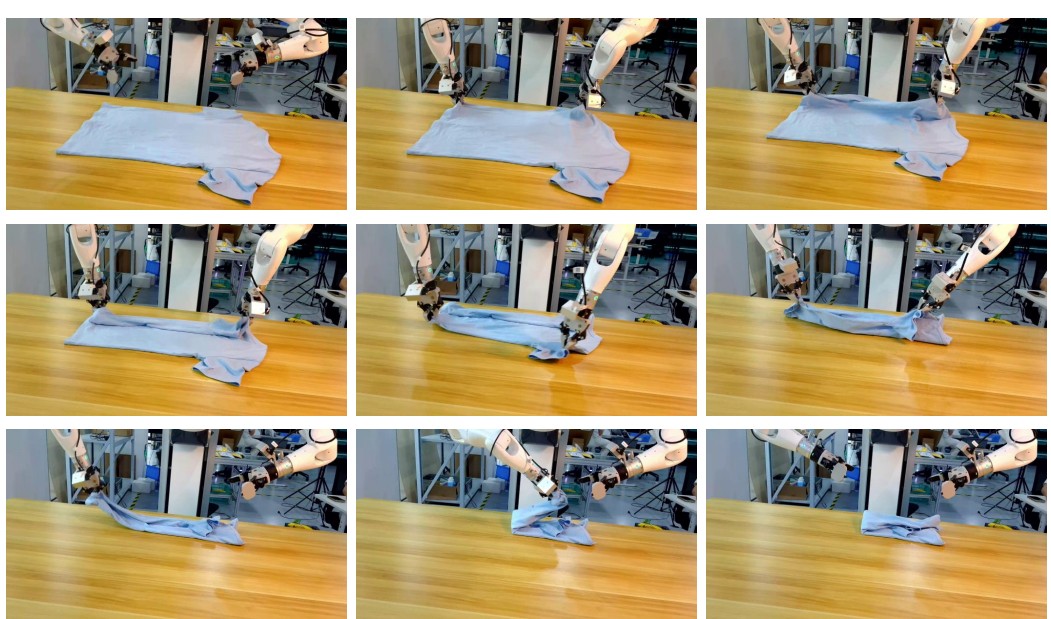

Figure 27: Robustness to texture and orientation. Successful execution with a back-oriented, blue cloth (size 160) on a textured wooden surface.

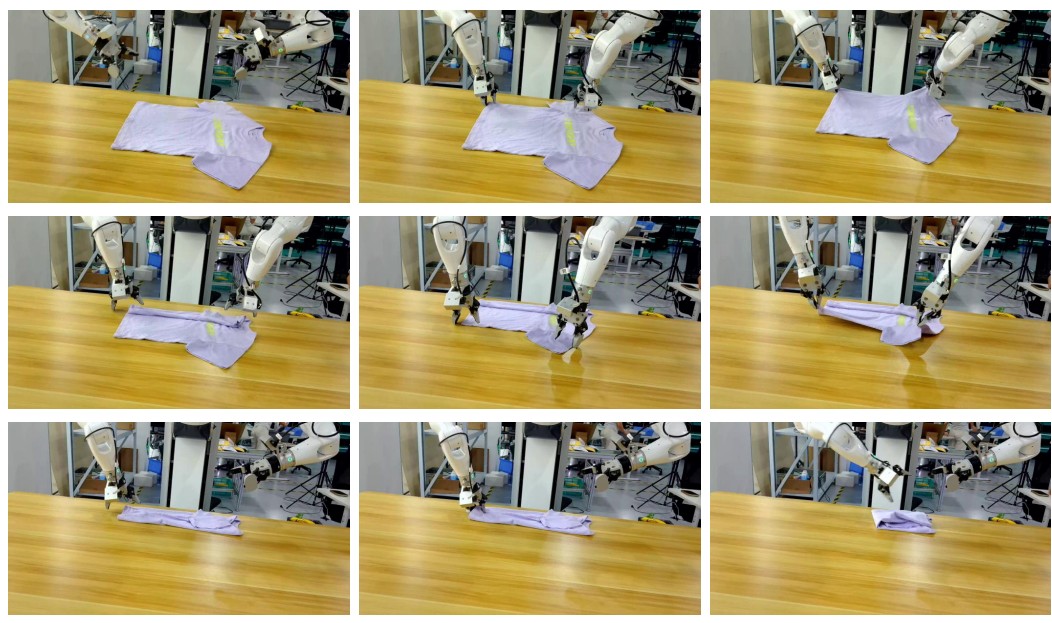

Figure 28: Generalization to novel object size. Successful execution with a front-oriented, purple cloth (size 120) on a wooden table surface.

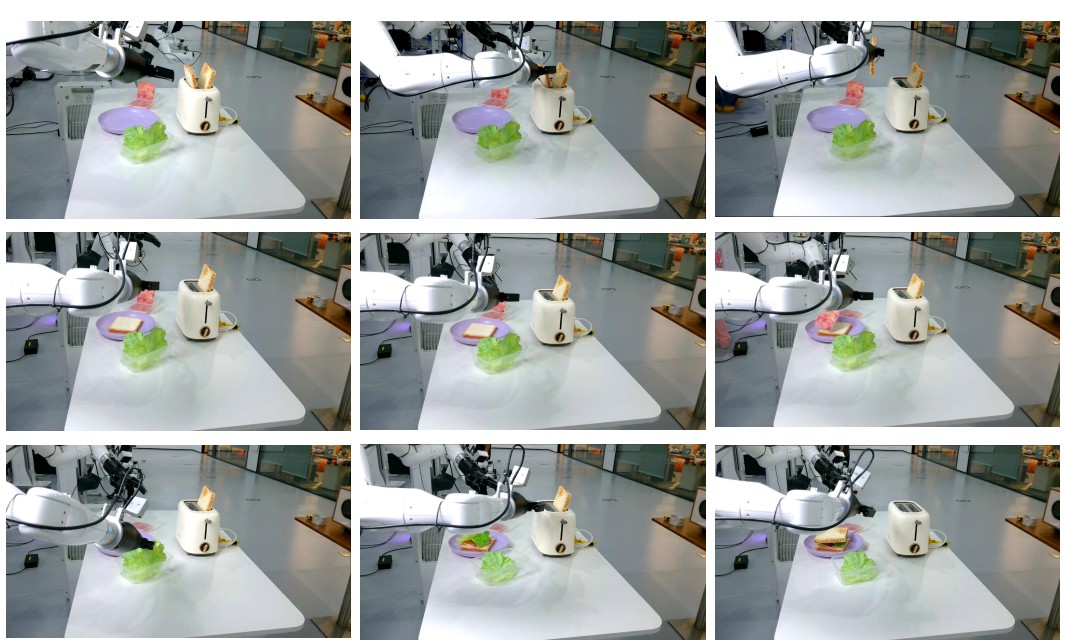

Figure 29: Dual-arm sandwich making. The policy layers deformable ingredients (bread, meat, lettuce) despite variable surface friction and partial occlusions.

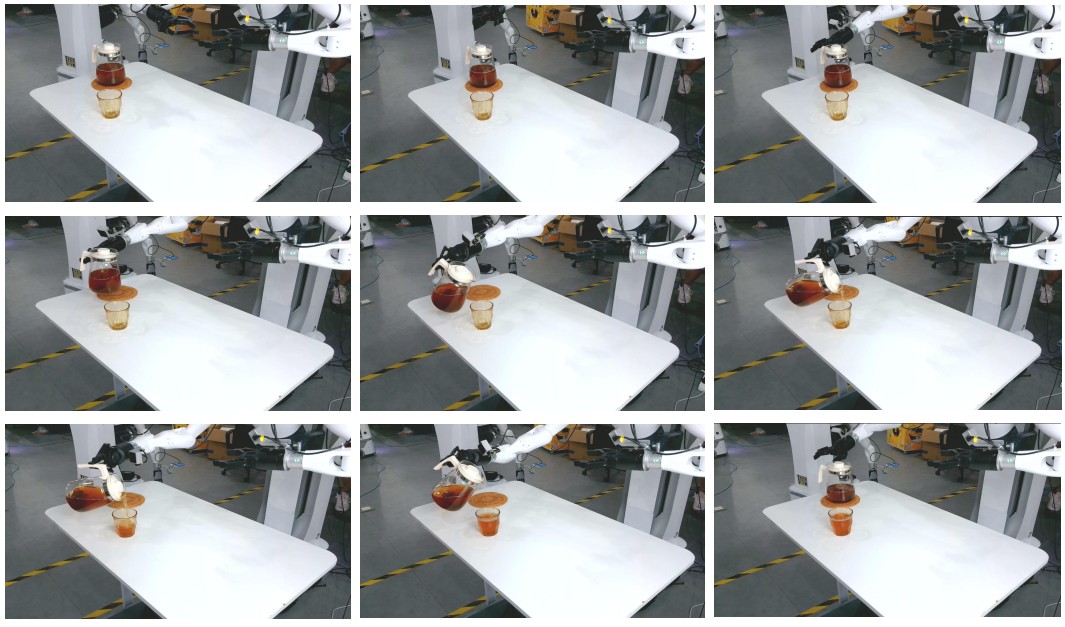

Figure 30: Single-arm controlled pouring. The policy regulates pouring angle and flow rate to transfer liquid to a target vessel.

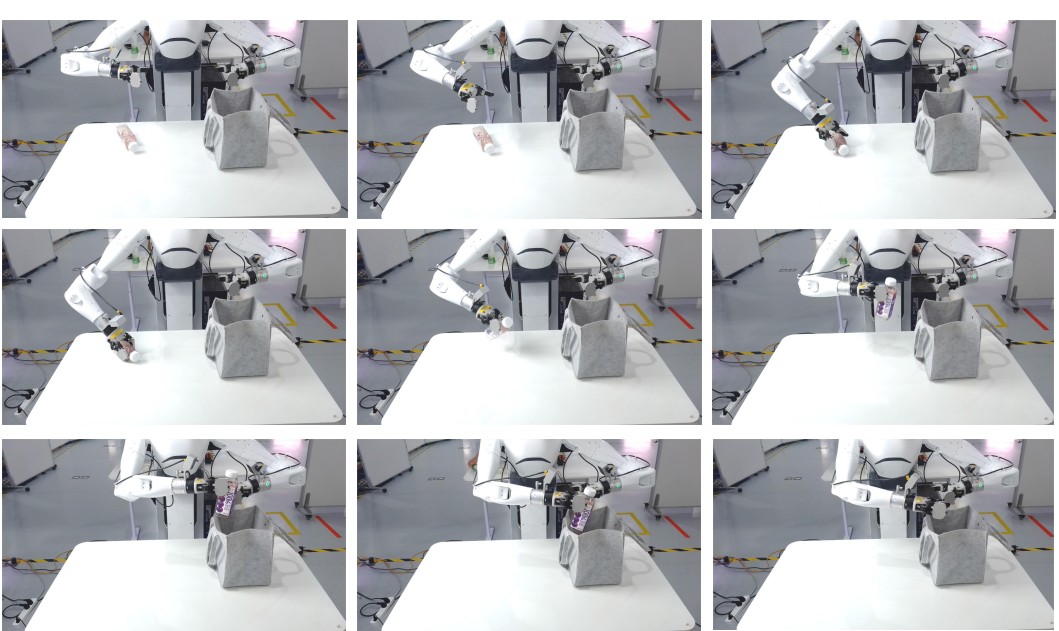

Figure 31: Orientation-aware bottle packing. The policy identifies object orientation to ensure stable placement within the container.

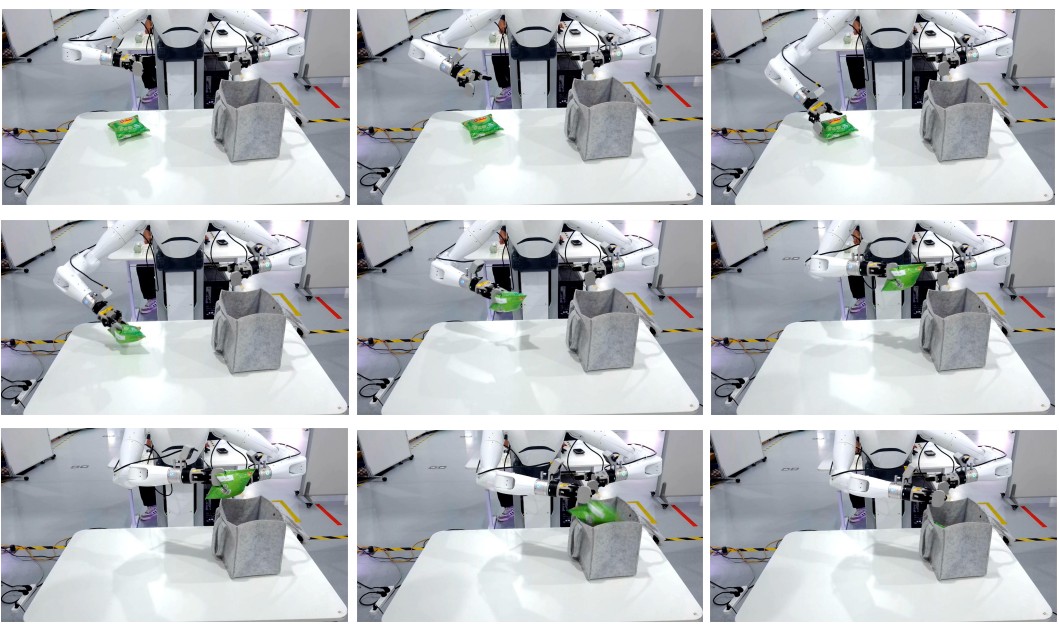

Figure 32: Texture-robust crisps packing. The policy generalizes across object textures and container shapes.

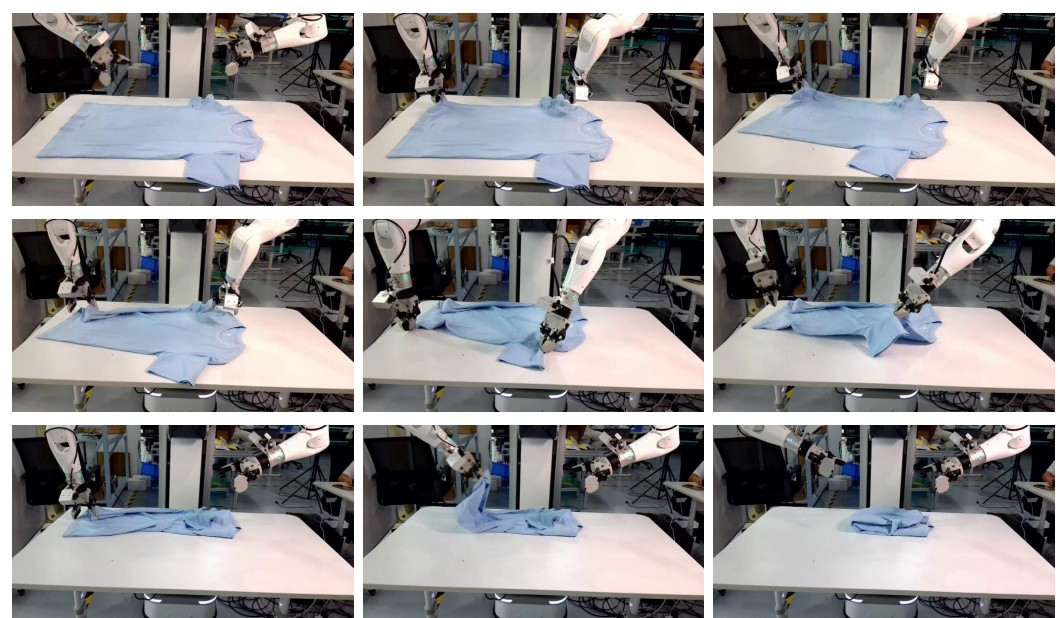

Figure 33: Failure case analysis. A trial where gripper jitter disrupted the trajectory during the folding phase, preventing task completion.

