# OpenReview forum: "Fidelity-Aware Data Composition for Robust Robot Generalization"
_ICLR.cc/2026/Conference — Submitted to ICLR 2026_

### Official Review · Reviewer_j786 · 2025-10-16

**Soundness:** 1
**Presentation:** 2
**Contribution:** 2
**Rating:** 2
**Confidence:** 4

**Summary:**

This paper addresses the problem of "shortcut learning" in robot policies, where models trained on visually homogeneous data fail to generalize to out-of-distribution (OOD) environments. The authors posit that the solution lies not just in generative data augmentation but in the principled composition of real and synthetic data. They introduce a framework, Coherent Information Fidelity Tuning (CIFT), which formalizes this composition as an optimization problem. The framework consists of two main components: 1) MVAug, a multi-view video-to-video generative model for creating a diverse spectrum of disentangled data, and 2) a composition algorithm that uses a feature-space Signal-to-Noise Ratio (SNR) as a proxy for "Information Fidelity." By analyzing this SNR, CIFT aims to identify an optimal mixing ratio (λ) of real-to-synthetic data that maximizes robustness while avoiding a "Decoherence Point" where training stability collapses. The authors apply CIFT to train π₀ and Diffusion Policy, reporting significant improvements (over 54%) in OOD task success rates on physical robots.

**Strengths:**

- The paper correctly identifies a critical and subtle challenge in modern robot learning. Moving the focus from just data synthesis to principled data composition is a valuable contribution to the field. The concept of a "Diversity-Information Fidelity trade-off" is insightful and well-articulated.
- The attempt to create a formal, data-driven framework (CIFT) to select the data mixing ratio is commendable. It moves beyond the common practice of ad-hoc hyperparameter tuning and seeks a more systematic solution.
- The authors present a complete system, from a sophisticated generative model (MVAug) to a composition algorithm, and validate it with closed-loop experiments on physical robots. The reported OOD performance improvements are substantial and demonstrate the potential of the underlying idea.

**Weaknesses:**

Despite its promising direction, the paper is undermined by significant methodological flaws, a lack of critical experimental validation, and poor clarity in key areas. The claims made are not sufficiently supported by the evidence provided.
- The central hypothesis is that CIFT's SNR proxy can identify an optimal mixing ratio λ* for closed-loop policy performance. However, the experiments fail to validate this.
1. The validation of the SNR proxy is done entirely in an open-loop setting (Section 5.2, Figure 5), where the metric of success is the "Robustness Score (RS)," an unvalidated metric based on action prediction MSE. It is well-known in robotics that open-loop performance (i.e., imitation accuracy on a static dataset) is a poor predictor of closed-loop performance, where compounding errors and unexpected states dominate.
2. The on-robot, closed-loop experiments (Table 3) only compare the baseline policy (λ=0, "w/o CIFT") against the policy trained with the CIFT-selected ratio (λ=λ*, "w/ CIFT"). Crucially, they do not evaluate other mixing ratios (e.g., λ values around the supposed "Decoherence Point") in the closed-loop setting. Without this comparison, the authors have only shown that their specific data augmentation is beneficial. They have provided no evidence that their CIFT selection procedure finds a better λ than any other randomly chosen λ, or that the "Decoherence Point" identified via open-loop analysis corresponds to a real collapse in closed-loop task success. This is the single most critical experiment needed to validate the paper's core contribution, and its absence is a fatal flaw.

- The paper is heavily motivated by the goal of solving shortcut learning (i.e., reliance on spurious features). However, it provides no direct evidence that the method actually reduces this phenomenon. Showing improved OOD generalization is not direct proof of it. The authors should have included experiments to investigate what the baseline vs. CIFT-trained policies are paying attention to in ID vs. OOD settings. Alternatively, they could have constructed controlled experiments where a specific spurious cue is present or absent to directly measure the policy's reliance on it. Without this, the claims about how the method works are purely speculative.

- Weak justification and opacity of proposed methods:
1. The paper provides insufficient detail on how the MVAug video-to-video model was trained (Appendix B). Crucially, it omits the dataset used for fine-tuning the Cosmos-Predict2-2B-Video2World foundation model. What data was used for paired multi-view video data? Without this information, the results are not reproducible, and it is impossible to assess whether the generative model's quality unfairly advantages their method over baselines.
2. The justification for using the SNR of the first principal component of features from a generic, pre-trained Inception-v3 model is weak and poorly explained. The text simply states, "Analysis shows a non-monotonic relationship..." but provides no citation or pointer to where this analysis is. Why should the axis of maximum variance in a general-purpose feature space be the most informative signal for the training dynamics of a specific robot policy? It is an unsubstantiated leap of faith.

- The authors report that training a Diffusion Policy required "approximately 80 hours on 16 H100 GPUs." This is an astronomical amount of computation for fine-tuning on a small dataset of 200 real episodes. This implies that the CIFT-composed dataset is enormous, meaning MVAug must generate a massive volume of synthetic data. This is not just a resource issue; it fundamentally questions the method's efficiency and practicality. It suggests the policy's robustness comes from brute-force exposure to an immense quantity of generated data, rather than a finely-tuned "optimal" composition. The paper should be more transparent about the scale of data generation required.

**Questions:**

- The core validation experiment (Figure 5, Table 1) shows that the peak Robustness Score (RS) occurs at a 100:200 ratio, while the CIFT method selects the 100:100 ratio based on peak SNR. Why should one trust a proxy (SNR) that selects a demonstrably sub-optimal point according to the open-loop evaluation metric (RS)?
- Could you elaborate on the justification for using the first principal component of Inception-v3 features as a proxy for information fidelity? A policy's learned representation is task- and architecture-specific. Why would the primary axis of variance from a generic, pre-trained image classifier be a reliable indicator of the complex learning dynamics and gradient alignment for a visuomotor policy like π₀?

---

> ### Author Response · Authors · 2025-11-23
> **3️⃣ Author Response to Reviewer j786 (Weakness 4, Questions, Refs)**
>
> >**Weakness 4**: Computational Efficiency and Reporting.
>
> **Response:**
> We apologize for a notation error in the initial draft where we mistakenly cited the original Diffusion Policy (Chi et al., 2023). By referring to our model broadly as a "diffusion policy," we inadvertently obscured the scale of the architecture. To clarify, we employed GE-Act (Liao et al., 2025), which is a policy model based on a diffusion process with 1.8B parameters, rather than the standard lightweight implementation.
>
> We have corrected a reporting error regarding the training duration. The initial estimate erroneously included data pre-processing time. The revised figures below reflect the pure fine-tuning latency strictly associated with the policy optimization on 8 NVIDIA H100 GPUs:
>
> | Model Architecture | Parameter Count | Actual Fine-Tuning Time |
> | :--- | :--- | :--- |
> | $\\pi_0$ **Foundation Model** (Black et al., 2024) | 3.3B | **~50 Hours** |
> | **GE-Act** (Liao et al., 2025) | 1.8B | **~24 Hours** |
>
> These figures confirm that the method is computationally feasible and consistent with standard foundation model fine-tuning. The robustness gains come from the quality of the CIFT-selected data composition, not brute-force scaling.
>
> ---
>
> >**Question 1**: Discrepancy between RS and SNR Peaks.
>
> **Response:**
> This observation highlights exactly why open-loop metrics are insufficient. As shown in our new data (Table 1 above), the Robustness Score (RS) peaked at the $100:200$ ratio, while the SNR peaked at $100:100$.
> Our physical validation confirmed that the $100:100$ ratio (selected by SNR) yielded the highest real-world success rate ($81.0\\%$), whereas the $100:200$ ratio dropped to $63.0\\%$. This empirical evidence proves that Feature-Space SNR is a more precise predictor of closed-loop optimality than open-loop error metrics like RS, which may not fully account for covariate shift.
>
> ---
>
> >**Question 2**: Elaboration on Inception-v3 Principal Component.
>
> **Response:**
> As detailed in our response to Weakness 3.2, we rely on the **Manifold Hypothesis** (Fefferman et al., 2016). Valid demonstrations occupy a lower-dimensional manifold, and the first principal component ($w_1$) captures the dominant causal signal (e.g., kinematic evolution). We utilize Inception-v3 because its supervised training on textures and shapes provides a stable basis for detecting when synthetic artifacts (noise) begin to dominate this principal signal. We mathematically modeled this phenomenon as **Spectral Stability** in **Proposition 4.1**, where the SNR acts as a tractable proxy for the alignment between the data variance and the task manifold.
>
> ---
>
> **References**
>
> * [**Black et al., 2024**] Kevin Black, Noah Brown, Danny Driess, et al. $\pi_0$: A vision-language-action flow model for general robot control. arXiv preprint arXiv:2410.24164, 2024.
> * [**Bu et al., 2025**] Qingwen Bu, Jisong Cai, Li Chen, et al. Agibot world colosseo: A large-scale manipulation platform for scalable and intelligent embodied systems. In IEEE/RSJ International Conference on Intelligent Robots and Systems (IROS), 2025.
> * [**Davis & Kahan, 1970**] Chandler Davis and William Morton Kahan. The rotation of eigenvectors by a perturbation. iii. SIAM Journal on Numerical Analysis, 7(1):1–46, 1970.
> * [**Fefferman et al., 2016**] Charles Fefferman, Sanjoy Mitter, and Hariharan Narayanan. Testing the manifold hypothesis. Journal of the American Mathematical Society, 29(4):983–1049, 2016.
> * [**Liao et al., 2025**] Yue Liao, Pengfei Zhou, Siyuan Huang, et al. Genie envisioner: A unified world foundation platform for robotic manipulation. arXiv preprint arXiv:2508.05635, 2025.
> * [**Chi et al., 2023**] Cheng Chi, Zhenjia Xu, Siyuan Feng, et al. Diffusion policy: Visuomotor policy learning via action diffusion. The International Journal of Robotics Research, 2023.

---

> ### Author Response · Authors · 2025-11-23
> **2️⃣ Author Response to Reviewer j786(Weaknesses 2 & 3)**
>
> >**Weakness 2**: Direct Evidence of Shortcut Mitigation.
>
> **Response:**
> We agree that general OOD generalization is not direct proof of shortcut removal. To address this, we designed and executed a "Trap" experiment (Section 5.4) to specifically isolate the policy's reliance on spurious cues ($v$) versus core causal features ($u$).
>
> **1. Experimental Design (Establishing Spurious Correlations)**
> We constructed biased datasets for two distinct tasks to embed strong spurious correlations ($v \rightarrow a$) that contradict the causal logic ($u \rightarrow a$):
> * **Cloth Folding:** We created a dataset where "Lighting On" was perfectly correlated with "Front-Facing Cloth" (mechanically requiring a Right-Side Fold).
> * **Toy Picking:** We created a dataset where a "White Table" was perfectly correlated with the "Object located on the Right" (requiring a Rightward Move).
>
> **2. The Trap (Inverting the Correlations)**
> We evaluated the policies in "Trap" settings that inverted these relationships to test if the policy would follow the visual shortcut ($v$) or the true state ($u$):
> * **Folding Trap:** Presented a "Back-Facing Cloth" (requires Left-Side Fold) under "Lighting On" conditions.
> * **Picking Trap:** Placed the object on the "Left Side" of a "White Table".
>
> **3. Results**
>
> | Task | Policy | Behavior in "Trap" Setting | Success Rate | Interpretation |
> | :--- | :--- | :--- | :--- | :--- |
> | Cloth Folding | Baseline | Folded Right Sleeve (Incorrect) | 5% | Relied on Lighting ($v$) |
> | **Cloth Folding** | **CIFT** | **Folded Left Sleeve (Correct)** | **75%** | **Relied on Cloth State ($u$)** |
> | Toy Picking | Baseline | Moved Right (Empty Space) | 0% | Relied on Table Color ($v$) |
> | **Toy Picking** | **CIFT** | **Moved Left (Towards Object)** | **80%** | **Relied on Object Pos. ($u$)** |
>
> The results confirm that the baseline policies were completely misled by the environmental shortcuts, while CIFT-trained policies correctly identified the causal structure.
>
> ---
>
> >**Weakness 3**: Methodological Details and Justification.
>
> **Response to 3.1: MVAug Training and Reproducibility**
> We have updated **Appendix B** to ensure full reproducibility.
> * **Dataset:** We fine-tuned the foundation model on **AgiBot-World-Beta** (Bu et al., 2025). This is a publicly available, large-scale benchmark containing over one million instruction-aligned multi-view video sequences (2967 hours). We highlight that the Alpha version of this dataset serves as a primary data source (accounting for ~80% of real-robot trajectories) for the state-of-the-art NVIDIA GR00T N1 foundation model (Bjorck et al., 2025). Using a standard open-source dataset ensures our results are comparable and not driven by proprietary data advantages.
> * **Training Protocol:** We have provided all necessary hyperparameters (e.g., 100,000 training steps, 8-bit AdamW optimizer, flow-matching objective) in **Table 3**. The performance gains stem from MVAug's specific architectural ability to maintain temporal and multi-view consistency via **Periodic Cross-View Attention**, rather than data secrecy.
>
> **Response to 3.2: Theoretical Justification for Feature-Space SNR**
> We have moved the theoretical derivation to the main text (**Section 4.2**) to substantiate our choice of metric.
>
> * **Spectral Stability:** We formally defined the "Decoherence Point" using **Proposition 4.1**, based on the **Davis-Kahan sin theorem** (Davis & Kahan, 1970). We proved that when the synthetic noise variance ($\Sigma_{noise}$) approaches the spectral gap ($\Delta$) of the manifold, the angle between the angle between the principal eigenvectors diverges:
>
> $$
> \sin\angle(w_1, v_1) \leq \frac{\lambda \Vert \Sigma_{\text{noise}} \Vert_2}{\Delta - \lambda \Vert \Sigma_{\text{noise}} \Vert_2}
> $$
>
> This signifies a geometric collapse where the feature signal decouples from the task structure.
>
> * **Backbone Selection:** Our ablation studies (**Figure 3**) demonstrate that **Inception-v3** is the most reliable observer for physical manipulation. Unlike CLIP (which saturates due to low semantic diversity in robotic instructions) or DINOv2 (which exhibited feature collapse in sparse environments), Inception-v3 captures the texture and shape fidelity required for detecting this specific geometric collapse.

---

> ### Author Response · Authors · 2025-11-23
> **1️⃣ Author Response to Reviewer j786 (General, Weakness 1)**
>
> We sincerely thank the reviewer for their rigorous and insightful feedback. We appreciate the identification of the gap regarding closed-loop validation and the need for direct evidence of shortcut mitigation. In response, we have significantly expanded our physical experiments, added a controlled "Trap" setting, and formalized the theoretical justification for our metric.
>
> >**Weakness 1**: Validation of SNR Proxy and Optimal Mixing Ratio ($\lambda^*$).
>
> **Response to 1.1 & 1.2 (Combined): Comprehensive Closed-Loop Physical Validation**
> We agree that open-loop metrics (like RS) are often insufficient predictors of robotic performance due to compounding errors. To address this, we moved beyond open-loop analysis and conducted extensive physical closed-loop evaluations.
>
> **1. Granular Physical Ablation**
> As requested, we did not merely compare the baseline ($\lambda=0$) with the optimal point. We trained and evaluated distinct policies for a sweep of mixing ratios ($100:0, 100:100, 100:200, 100:300$, etc.) on the physical robot.
>
> **2. Empirical Confirmation of the Decoherence Point**
> The results, summarized in the table below, explicitly validate the SNR proxy:
>
> | Mixing Ratio (Real:Synth) | Feature-Space SNR ($\vert \mu/\sigma \vert$) | Robustness Score (RS) | Real-World Success Rate (%) | Status |
> | :--- | :--- | :--- | :--- | :--- |
> | 100:0 (Baseline) | 0.1423 | 0.00 | 35.00 | Reference |
> | **100:100 (CIFT Choice)** | **0.2171 (Peak)** | 56.37 | **81.0 (Peak)** | **Optimal** |
> | 100:200 | 0.1644 | **60.29 (Peak)** | 63.0 | Sub-optimal |
> | 100:300 | 0.0097 (Drop) | 36.56 (Drop) | 34.0 (Collapse) | Decoherence |
> | 100:400 | 0.0588 | 53.91 | 40.0 | Unstable |
> | 100:500 | 0.1448 | 41.54 | 45.0 | Unstable |
>
> * **Optimal Point Alignment:** The Feature-Space SNR peaked at the $100:100$ ratio ($|\mu/\sigma|=0.2171$). Crucially, this ratio achieved the highest Real-World Success Rate ($81.0\\%$), outperforming both the baseline ($35.0\\%$) and the ratio favored by the open-loop RS metric ($100:200$).
> * **Decoherence Point Verification:** The SNR metric sharply declined at the $100:300$ ratio (dropping to $0.0097$). Physical testing confirmed this collapse, with the success rate plummeting to $34.0\\%$ (comparable to the baseline).
>
> This granular analysis provides direct evidence that the CIFT selection procedure ($\lambda^*$) identifies the optimal operating point for closed-loop control better than random selection, and that the "Decoherence Point" corresponds to a real-world performance collapse.

---

### Official Review · Reviewer_X2Vq · 2025-10-29

**Soundness:** 3
**Presentation:** 3
**Contribution:** 3
**Rating:** 6
**Confidence:** 2

**Summary:**

The paper introduces **Coherent Information Fidelity Tuning (CIFT)**, a new framework to optimize the composition of real and synthetic data for robot policy training. CIFT consists of two major components: Multi-View Video Augmentation (MVAug) — a latent diffusion transformer that generates multi-view, causally consistent video demonstrations; Information Fidelity Metric — a quantitative proxy for data quality that measures coherence between real and synthetic data using the feature-space signal-to-noise ratio. Empirical evaluations on real-world robotic manipulation tasks show that CIFT achieves superior performance over existing baselines.

**Strengths:**

- Using SNR to quantitatively measures the coherence is both interesting and novel.
- The authors conduct comprehensive experiments, including extensive ablation studies, demonstrating the effectiveness of CIFT.
- The Robustness Score of MVAug surpasses that of the baseline methods by a large margin across all mixing ratios, demonstrating the superiority of the generated data.
- The proposed method can be integrated with various pretrained backbones or downstream algorithms, making it broadly applicable across different robot learning setups.

**Weaknesses:**

- The Feature SNR used as a proxy for robustness does not appear to align well with the actual robustness score, as shown in Figure 5(a).
- The authors state that all code and datasets have been made publicly available in an anonymous repository. However, I was unable to locate or access this repository. I only found that some video demonstrations are included in the supplementary material.

**Questions:**

- It is interesting to observe that quite different pretrained backbones (Inception-v3, CLIP, DINO-v2) yield exactly the same Decoherence Point. Could the authors provide an explanation for why this happens?
- Could the authors clarify how to access the code and dataset mentioned in the paper?

---

> ### Author Response · Authors · 2025-11-23
> **2️⃣ Author Response to Reviewer X2Vq (Weakness 2, Questions, Refs)**
>
> >**Weakness 2 & Question 2**: Repository Access.
>
> **Response:**
> We sincerely apologize for the oversight in our initial submission regarding the visibility of the repository link. We have updated the Abstract and Introduction of the revised manuscript to explicitly include the link to the anonymous repository:
>
> **Link:** https://anonymous.4open.science/r/CIFT-code
>
> Furthermore, noting that this anonymous hosting platform occasionally experiences intermittent connectivity issues, we have also included the full source code archive in the Supplementary Material to ensure reliable access.
>
> We confirm that this repository is fully populated and provides the core materials necessary to reproduce our methodological contributions:
> * **CIFT Framework:** Complete implementation of the Coherent Information Fidelity Tuning algorithms, including the Feature-Space SNR calculation (Section 4.2).
> * **MVAug Engine:** Source code for both the training and inference of the Multi-View Video Augmentation generative model (Section 4.1; Appendix B).
> * **Datasets & Processing:** Access to the datasets used in this study and associated data processing scripts (e.g., AgiBot-World-Beta (Bu et al., 2025), a core component of the NVIDIA GR00T N1 pre-training corpus (Bjorck et al., 2025)).
>
> As stated in our **Reproducibility Statement**, we are fully committed to open science. Upon the conclusion of the review process, we will release a public repository containing all code, datasets, and pre-trained model checkpoints.
>
> ---
>
> >**Question 1**: Consistency of Pre-trained Backbones (Inception-v3, CLIP, DINOv2).
>
> **Response:**
> We thank the reviewer for this interesting observation. We would like to clarify that while the backbones exhibit similar trends in the high-complexity "Folding Clothes" task (as seen in Figure 3a), they do not consistently identify the same Decoherence Point across all tasks.
> As illustrated in **Figure 3(b)** and discussed in **Section 4.2**, significant divergences appear in simpler tasks (e.g., "Wiping a Table"). These differences reveal critical interactions between the model architectures and the robotic data distribution:
>
> | Backbone | Behavior in Simple Tasks (e.g., Wiping) | Underlying Mechanism |
> | :--- | :--- | :--- |
> | **CLIP** (Radford et al., 2021) | High Volatility | **Semantic Saturation:** CLIP relies on text-image alignment. Robotic instructions are often repetitive (e.g., "wipe table"), causing saturation in the latent space. The metric becomes sensitive to noise rather than structure. |
> | **DINOv2** (Oquab et al., 2024) | Feature Collapse | **Object-Centric Bias:** DINOv2 is optimized for object-centric features. In sparse robotic environments with static objects, it struggles to extract meaningful variance, leading to near-zero signal strength. |
> | **Inception-v3** (Szegedy et al., 2016) | **Stable Detection** | **Texture/Shape Bias:** Trained on supervised classification, Inception-v3 retains a strong bias towards visual textures and shapes. This aligns well with physical manipulation variations, providing the most consistent measurement of geometric collapse. |
>
> **Conclusion:**
> While all models may detect signal breakdown in highly complex scenes, only **Inception-v3** reliably identifies the Decoherence Point across the full spectrum of task complexities. This robustness motivated our selection of Inception-v3 as the standard backbone for the CIFT framework.
>
> **References**
>
> * [**Bu et al., 2025**] Qingwen Bu, Jisong Cai, Li Chen, et al. Agibot world colosseo: A large-scale manipulation platform for scalable and intelligent embodied systems. In IEEE/RSJ International Conference on Intelligent Robots and Systems (IROS), 2025.
> * [**Oquab et al., 2024**] Maxime Oquab, Timothée Darcet, Théo Moutakanni, et al. Dinov2: Learning robust visual features without supervision. Transactions on Machine Learning Research Journal, pp. 1-31, 2024.
> * [**Radford et al., 2021**] Alec Radford, Jong Wook Kim, Chris Hallacy, et al. Learning transferable visual models from natural language supervision. In International Conference on Machine Learning (ICML), 2021.
> * [**Szegedy et al., 2016**] Christian Szegedy, Vincent Vanhoucke, Sergey Ioffe, Jon Shlens, and Zbigniew Wojna. Rethinking the inception architecture for computer vision. In Proceedings of the IEEE/CVF Conference on Computer Vision and Pattern Recognition (CVPR), 2016.

---

> ### Author Response · Authors · 2025-11-23
> **1️⃣ Author Response to Reviewer X2Vq (Weakness 1)**
>
> >**Weakness 1**: Alignment between Feature-Space SNR and Robustness Score (RS).
>
> **Response:**
> We thank the reviewer for this keen observation regarding Figure 5(a). We acknowledge that in our initial open-loop analysis, the peak of the Feature-Space SNR (at 100:100) did not perfectly align with the peak of the Robustness Score (RS) (at 100:200). We address this divergence below:
>
> **1. Limitations of Open-Loop RS as a Ground Truth**
> As discussed in the revised manuscript, open-loop metrics like RS (derived from trajectory prediction MSE) are useful heuristics but suffer from the "covariate shift" problem inherent in robotic control. Small open-loop errors often compound into failures during long-horizon execution. Therefore, RS should be viewed as a directional indicator rather than an absolute ground truth for policy performance.
>
> **2. SNR Aligns Best with Real-World Success (Ground Truth)**
> To resolve this ambiguity, we conducted extensive real-world physical evaluations to establish the true performance landscape. As detailed in the updated **Table 1**, the physical results demonstrate that Feature-Space SNR is a significantly more accurate predictor of closed-loop success than RS:
>
> | Mixing Ratio (Real:Synth) | Feature-Space SNR ($\vert \mu/\sigma \vert$) | Robustness Score (RS) | Real-World Success Rate (%) | Conclusion |
> | :--- | :--- | :--- | :--- | :--- |
> | **100:100 (CIFT Choice)** | **0.2171 (Peak)** | 56.37 | **81.0 (Peak)** | **SNR correctly identifies the Optimal Point** |
> | 100:200 | 0.1644 | **60.29 (Peak)** | 63.0 | RS overestimates performance here |
> | 100:300 | 0.0097 (Drop) | 36.56 (Drop) | 34.0 (Collapse) | All metrics identify the Decoherence Point |
>
> * **Optimal Point Alignment:** The SNR metric peaks at the $100:100$ ratio. Crucially, this exact ratio corresponds to the peak Real-World Success Rate ($81.0\\%$). In contrast, while RS peaks at $100:200$, the actual physical success rate at that ratio drops significantly to $63.0\\%$.
> * **Decoherence Point Consensus:** All metrics—SNR, RS, and Physical Success—uniformly identify the $100:300$ ratio as the point of collapse (Success Rate drops to $34.0\\%$, comparable to the baseline).
>
> **Conclusion:**
> These physical experiments confirm that Feature-Space SNR is a highly reliable proxy for the true objective—closed-loop robot performance. It correctly identified the optimal data composition ratio ($100:100$) where the policy achieved its maximum generalization capability.

---

### Official Review · Reviewer_4SFU · 2025-10-31

**Soundness:** 3
**Presentation:** 2
**Contribution:** 4
**Rating:** 6
**Confidence:** 3

**Summary:**

The paper proposes a unified framework that balances data diversity and realism in robot learning by combining Multi-View Video Augmentation (MVAug) and Coherent Information Fidelity Tuning (CIFT). MVAug is a latent diffusion-based generator that produces physically consistent and multi-view coherent augmented videos through cross-view attention and structure/appearance conditioning, enriching visual diversity without breaking causality. CIFT then analyzes the feature-space Signal-to-Noise Ratio (SNR) to quantify data fidelity and identify the optimal real-to-synthetic mixing ratio. Experiments on simulated and real robots demonstrate that the fidelity-aware composition strategy enhances policy robustness and generalization under unseen visual conditions, though its scalability and reliance on generative model quality remain limitations.

**Strengths:**

1. The paper proposes a well-founded framework CIFT that balances data diversity and realism. It also helps researchers predict the optimal real/synthetic data ratio without post-training, improving policy generalization and lowering computational cost.
2. This paper establishes a measurable relationship between feature-space SNR and robustness score (RS). SNR provides an interpretable and computationally efficient proxy for assessing data quality before training.
3. The latent diffusion transformer with periodic cross-view attention and structure/appearance conditioning achieves impressive results in maintaining geometric and temporal consistency across views.

**Weaknesses:**

1. The “decoherence point” $\lambda_{dc}$ is detected empirically from SNR minima, but this paper does not provide a rigorous mathematical criterion or theoretical justification for its general validity.
2. The experimental results are based on a single checkpoint with 20 trials and limited task types, making the improvement in Table 3 less reliable. Additional policy testing results from tasks in MVAug synthesis examples (in Appendix) and details on settings, such as the object's initial position range, should be provided.

**Questions:**

1. What is the computational overhead (training/inference time) for generating augmented datasets using MVAug?
2. A minor suggestion: in Figure 23, please consider changing “bottom” and “top” to “left” and “right” for clarity.

---

> ### Author Response · Authors · 2025-11-23
> **3️⃣ Author Response to Reviewer 4SFU (Questions 1-2, Refs)**
>
> >**Question 1:** Computational Overhead
>
> **Response:**
> We have included a detailed breakdown of the computational overhead in **Appendix B** and **Table 4**.
>
> **1. Data Generation Efficiency (MVAug)**
> The generation process is efficient on consumer hardware (NVIDIA RTX 4090):
> * **MVAug (Ours):** 2.38 FPS. This is approximately **5-7x faster** than RoboTransfer (0.35 FPS).
> * **RoboTransfer** (Liu et al., 2025): 0.35 FPS. The baseline is significantly slower and prone to Out-Of-Memory (OOM) errors on long sequences.
>
> **2. Policy Training Context**
> We fine-tuned the $\\pi_0$ model for approximately 50 hours on 8 NVIDIA H100 GPUs.
> * **Context:** This cost reflects the inherent complexity of bimanual manipulation with a 14-DoF action space. Learning precise coordination for tasks like cloth folding requires foundational VLA models, which naturally demand significant compute.
> * **Scalability:** For simpler single-arm tasks, the data requirements and training time would be proportionally lower. The overhead is driven by the task difficulty and model architecture, not an inefficiency in the CIFT method itself.
>
> ---
>
> >**Question 2:** Figure 23 Caption
>
> **Response:**
> We thank the reviewer for their attention to detail. We have updated the caption of Figure 23 in the revised manuscript to correctly refer to the "left" and "right" panels, ensuring the description accurately matches the visual layout.
>
> ---
>
> **References**
>
> * [**Davis & Kahan, 1970**] Chandler Davis and William Morton Kahan. The rotation of eigenvectors by a perturbation. iii. SIAM Journal on Numerical Analysis, 7(1):1–46, 1970.
> * [**Liu et al., 2025**] Liu Liu, Xiaofeng Wang, Guosheng Zhao, et al. Robotransfer: Geometry-consistent video diffusion for robotic visual policy transfer. arXiv preprint arXiv:2505.23171, 2025.
> * [**Team et al., 2025**] TRI LBM Team, Jose Barreiros, Andrew Beaulieu, et al. A careful examination of large behavior models for multitask dexterous manipulation. arXiv preprint arXiv:2507.05331, 2025.

---

> ### Author Response · Authors · 2025-11-23
> **2️⃣ Author Response to Reviewer 4SFU (Weakness 2)**
>
> >**Weakness 2:** Statistical Rigor and Task Diversity
>
> **Response:**
> We sincerely thank the reviewer for emphasizing the need for greater statistical rigor. We agree that a single checkpoint is insufficient. We have significantly expanded our evaluation protocol to ensure reliability.
>
> **1. Enhanced Statistical Rigor (3 Independent Runs)**
> We have moved beyond single-checkpoint evaluation. As detailed in Section 5.4, we now report the mean and standard deviation across **three independent training runs** (using different random seeds) for all physical experiments, following the protocol in (Team et al., 2025).
> * **Visual Evidence:** This variance is explicitly visualized as error bars in **Figure 7** and **Figure 8**. The tight error bars observed on the CIFT-trained policies confirm that the performance improvement is statistically significant and not a result of lucky initialization.
>
> **2. Expansion of Task Types**
> We have expanded our physical evaluation suite to cover five distinct manipulation tasks, demonstrating robustness across diverse dynamics:
>
> | Task Type | New Task Added | Challenge Addressed | Performance Note |
> | :--- | :--- | :--- | :--- |
> | **Deformable Objects** | Sandwich Making | Variable friction & deformation | Failure rate reduced from ~70% to ~40% under lighting shifts. |
> | **Fluid Dynamics** | Pouring Tea | Precision & fluid handling | Achieved success where baseline failed (<20%) due to object shifts. |
> | **Orientation** | Packing | Orientation-aware placement | Robust to unseen brands and continuous packing states. |
>
> **3. Detailed Experimental Settings**
> We have added **Appendix D.1** and **Table 9** to explicitly list the configuration details:
> * **Initialization:** We utilize "Randomized Initial Poses" for grasping and specific "canonical orientations" (front/back) for folding to rigorously test spatial generalization.
> * **Variations:** We explicitly list variations, such as "3 sizes and 2 colorways" for cloth and specific surface materials (leather, cotton, linen).

---

> ### Author Response · Authors · 2025-11-23
> **1️⃣ Author Response to Reviewer 4SFU (Weakness 1)**
>
> We sincerely thank the reviewer for their constructive feedback, particularly regarding the theoretical foundation of our metric and the statistical rigor of our experiments. We have significantly revised the manuscript to fully address these points.
>
> >**Weakness 1:** Theoretical Justification for the "Decoherence Point"
>
> **Response:**
> We thank the reviewer for highlighting the need for a rigorous theoretical foundation. We acknowledge that the initial version relied heavily on empirical observations. In response, we have established a formal mathematical criterion for the "Decoherence Point" in the revised manuscript.
>
> **1. Mathematical Definition via Spectral Stability (Section 4.2)**
> We have introduced **Proposition 4.1 (Spectral Stability)** in Section 4.2 to provide a rigorous derivation based on the perturbation theory of covariance matrices.
>
> * **The Criterion:** We model the data composition as a perturbation problem where synthetic data acts as a perturbation matrix $\Sigma_{noise}$ to the real data covariance $\Sigma_{real}$. Using the Davis-Kahan sin theorem (Davis & Kahan, 1970), we derive a bound on the rotation of the principal eigenvector (the task "signal" $w_1$) relative to the true causal direction ($v_1$): $$\sin\angle(w_1, v_1) \leq \frac{\lambda \Vert \Sigma_{\text{noise}} \Vert_2}{\Delta - \lambda \Vert \Sigma_{\text{noise}} \Vert_2}$$
> where $\Delta$ is the spectral gap of the real data.
>
> * **The Theoretical Decoherence Point:** This allows us to mathematically define the "Decoherence Point" not as a heuristic, but as the singularity where the perturbation magnitude approaches the spectral gap ($\lambda\|\Sigma_{noise}\|_{2} \to \Delta$). At this threshold, the denominator vanishes, implying that the principal direction of the data mixture can rotate arbitrarily, effectively decoupling from the underlying causal structure of the task.
>
> **2. Connection to Learning Dynamics**
> We have further linked this geometric collapse to optimization stability. Our analysis demonstrates that this feature-space collapse corresponds to a regime of **Gradient Interference**, where the cosine similarity between real and synthetic gradients becomes negative, actively hindering the learning process.
>
> **Conclusion:**
> Consequently, the SNR minimum observed in our experiments is a measurable proxy for this theoretically derived phase transition in the feature-space geometry.

---

### Official Review · Reviewer_n7iA · 2025-11-11

**Soundness:** 3
**Presentation:** 3
**Contribution:** 3
**Rating:** 4
**Confidence:** 3

**Summary:**

The paper proposes a multi-view data augmentation approach to achieve better policy performance when finetuned on a target domain and when tested especially under distribution shift. The author propose a multi-view consistent diffusion model that augments the robot demonstration data with different backgrounds, distractors, relighting and foreground objects etc. They propose a metric which guides how much to mix the original data with synthetic data. Both open and close loop experiments are performed with existing foundation models i.e. pi-0 as the base model.

**Strengths:**

In my opinion, here are the strengths of the paper:

1. A systematic study for robot learning foundation models on finding the feature space SNR correlates well with final post training policy stability.

2. Significant performance improvement vs the base model finetuning using the augmented dataset proposed by the paper.

3. The paper is clearly written and easy to follow, and figures complement the text nicely.

**Weaknesses:**

In my opinion, below are the weaknesses of the paper:

1. While it is great to study open-loop correlation, I am curious if the same correlation holds with the closed-loop SR. It is also not clear if open-loop RS score even correlates well with closed-loop SR; which makes the decisions on mixing ratio being calculated on open-loop RS seems a bit questionable.

2. While it is great to see qualitative results with baseline, I am wondering will the baselines i.e. RoboTransfer etc. would also result in the same performance improvements as the authors see in Table 3 with their proposed augmentations?

3. No statistical error bars are presented, which doesn't show the full variance b/w runs in these generative policies [1].

4. The qualitative results for RoboEngine are presented in a bit too extreme case where in every frame the background etc changes, is there a more principled baseline where one could take masked robot embodiment from RoboEngine and applies let's say same lightning to all frames or same color augmentation to foreground objects like shirt in all frames, where the results could look a lot better and cleaner for the baseline than currently presented.

5. Does the same augmentation strategy hold for pretraining the foundation model as opposed to finetuning where still 200 real-world demos are collected which is a large number.


[1] TRI LBM Team, A Careful Examination of Large Behavior Models for Multitask Dexterous Manipulation, arXiv 2025

**Questions:**

Please see questions in the weakness section. I look forward to seeing the author's response in rebuttal.

---

> ### Author Response · Authors · 2025-11-23
> **3️⃣ Author Response to Reviewer n7iA (Weaknesses 3, 5, Refs)**
>
> >**Weakness 3:** Statistical Significance & Reproducibility.
>
> **Response:**
> We thank the reviewer for raising this critical point. We agree that reporting statistical variance is essential for evaluating the reliability of generative policies, as highlighted in TRI LBM Team (2025).
>
> **1. Inclusion of Error Bars**
> We have updated our physical experimental results to include statistical error bars.
> * **Methodology:** As detailed in Section 5.4, we now report the standard deviation across three independent training runs to explicitly capture the variance in policy performance.
> * **Visualizations:** We have updated key result charts, including Figure 7 (Generalization performance on $\pi_0$) and Figure 8 (Ablation studies), to display these error bars.
>
> **2. Interpretation of Results**
> The inclusion of error bars confirms that our performance gains are statistically significant.
> * **Figure 7:** The error bars reveal that while baseline policies often exhibit high variance (instability) or consistently low performance under OOD conditions, the CIFT-augmented policies maintain high success rates with tight variance.
> * **Stability:** This validates our claim that CIFT not only improves average success rates but also enhances the training stability of the policy.
>
> ---
>
> >**Weakness 5:** Applicability to Pre-training & Data Efficiency (200 Demos)
>
> **Response:**
> We thank the reviewer for this forward-looking question. We clarify that our data collection strategy was designed to ensure a rigorous evaluation of robustness on highly complex tasks, rather than relying on brute-force data scaling.
>
> **1. Data Volume is Dictated by Task Complexity**
> We emphasize that the requisite data volume is a function of task difficulty relative to the foundation model's priors.
> * **High-Complexity Benchmarks:** We deliberately selected challenging, long-horizon tasks for evaluation, such as bimanual cloth folding and precision fluid pouring. For these high-DoF tasks, current foundation models (e.g., $\\pi_0$) typically require approximately 200 demonstrations to adapt effectively and achieve a stable, non-trivial success rate.
> * **Ensuring a Strong Baseline:** Our objective was to benchmark CIFT against a robust, capable baseline ($35\\%$ success in folding) rather than a weak one. Using fewer demonstrations for these complex tasks would likely result in near-zero baseline performance, obscuring the specific contribution of our augmentation strategy.
> * **Scalability:** For simpler tasks (e.g., pick-and-place), significantly fewer demonstrations would suffice to establish a baseline. CIFT acts as a scalable engine that enhances performance regardless of the base data requirement.
>
> **2. Integration into the VLA Training Pipeline**
> While our experiments focus on the "last mile" of post-training adaptation, these results have significant implications for the broader training pipeline.
> * **Foundation-Quality Data:** Notably, the dataset used in this study (AgiBot-World) is of foundation-model quality; its Alpha version specifically constitutes approximately 80% of the real-robot pre-training data for NVIDIA GR00T N1 (Bjorck et al., 2025; arXiv:2503.14734).
> * **Pre-training Implications:** Our method demonstrates that high-fidelity generative augmentation can bridge the gap between general pre-training and specific embodiment adaptation. This strategy holds promise for advanced paradigms like Iterative Offline Reinforcement Learning (e.g., $\pi_{0.6}$, Black et al., 2025), where CIFT can serve as a generative engine to expand the state space exploration of limited real-world datasets during large-scale training.
>
> ---
>
> **References**
>
> * [**Black et al., 2024**] Kevin Black, Noah Brown, Danny Driess, et al. $\pi_0$: A vision-language-action flow model for general robot control. arXiv preprint arXiv:2410.24164, 2024.
> * [**Black et al., 2025**] Kevin Black, Noah Brown, James Darpinian, et al. $\pi_{0.5}$: A vision-language-action model with open-world generalization. arXiv preprint arXiv:2504.16054, 2025.
> * [**Kuaishou, 2024**] Kuaishou. Klingai. [suspicious link removed], 2024.
> * [**Liu et al., 2025**] Liu Liu, Xiaofeng Wang, Guosheng Zhao, et al. Robotransfer: Geometry-consistent video diffusion for robotic visual policy transfer. arXiv preprint arXiv:2505.23171, 2025.
> * [**Team et al., 2025**] TRI LBM Team, Jose Barreiros, Andrew Beaulieu, et al. A careful examination of large behavior models for multitask dexterous manipulation. arXiv preprint arXiv:2507.05331, 2025.

---

> ### Author Response · Authors · 2025-11-23
> **2️⃣ Author Response to Reviewer n7iA (Weaknesses 2 & 4)**
>
> >**Weakness 2:** Performance of Baselines **Weakness 4:** Fairness of Baseline Implementation
>
> **Response to W2 & W4:**
> We thank the reviewer for these questions regarding the fairness of our baselines and the relationship between generative fidelity and policy robustness. We address both points together as they pertain to why high-fidelity generation is a prerequisite for successful policy learning.
>
> **1. Addressing the "Extreme Case" in RoboEngine (Response to W4)**
> The reviewer asks if a more "principled" application of RoboEngine (e.g., applying consistent lighting prompts to masked frames) would yield cleaner results.
> * **Inherent Architectural Limitation:** We clarify that the flickering artifacts observed in RoboEngine are not due to extreme prompting or varying conditions per frame. Rather, they are inherent to RoboEngine's architecture, which employs frame-by-frame background diffusion. Even when using a fixed prompt (e.g., "same lighting") and consistent masks, the independent diffusion process for each frame generates temporally incoherent backgrounds. This high-frequency visual jitter is quantitatively reflected in RoboEngine's high FVD score of 1463.5 (Table 2) compared to MVAug's 545.7.
> * **Baseline Implementation:** We optimized the baselines for our tri-view setting to the best of our ability. For RoboTransfer (Liu et al., 2025), the method relies on strict geometric conditions (depth/normal maps) and separated foreground/backgrounds. As discussed in Appendix C.3, this "cut-and-paste" synthesis strategy often produces unrealistic object contours, which act as high-frequency noise to the policy.
>
> **2. The Necessity of High Fidelity for Policy Learning (Response to W2)**
> The reviewer asks if baselines could achieve similar performance improvements. Our empirical results show they cannot, specifically due to the fidelity issues described above.
> * **Visual Artifacts as Noise:** In robotic learning, temporal inconsistency (flicker) and geometric artifacts are not merely cosmetic; they act as high-magnitude noise that corrupts the learning signal.
> * **Performance Gap:** Consequently, policies trained with RoboTransfer and RoboEngine degrade rapidly in our experiments (Figure 5c), with RS dropping to $\approx 20-35$, compared to MVAug (RS $\approx 60$). The policy effectively fails because it overfits to synthetic artifacts rather than learning robust physical representations.
>
> **3. Commercial Models and Hallucinations**
> To further investigate if "cleaner" video generation solves the issue, we experimented with KlingAI (Kuaishou, 2024) (a commercial SOTA video model) using reference image conditioning. While KlingAI produces high visual fidelity (sharp textures), it suffers from physical hallucinations (e.g., morphing table geometry or distorting robot arms). These hallucinations break the causal structure of the demonstrations, making the data detrimental to control policies despite looking "clean" to the human eye.
>
> **Conclusion:**
> MVAug was developed precisely because no existing method—open-source or commercial—could simultaneously satisfy the temporal consistency, multi-view geometry, and physical plausibility required for robotic policy learning.

---

> ### Author Response · Authors · 2025-11-23
> **1️⃣ Author Response to Reviewer n7iA (Weakness 1)**
>
> >**Weakness 1:** Metric Validity: Open-Loop RS vs. Closed-Loop SR
>
> **Response:**
> We thank the reviewer for this insightful comment. We agree that open-loop metrics are generally not perfect proxies for closed-loop success rates (SR) in robotic learning. However, in Section 5.2 and Section 5.3, we provide extensive empirical evidence validating that our **Robustness Score (RS)** is a reliable heuristic for two specific objectives: identifying the optimal data mixing ratio and detecting the critical failure boundary (which we term the "Decoherence Point").
>
> **1. Strong Correlation in Trends and Failure Points**
> While the absolute values of open-loop errors do not map linearly to success rates, the relative trends and inflection points are highly consistent between the two. As presented in Table 1 and Figure 5(a), our real-world experiments confirm this correlation:
> * **Consistency at the Peak:** The ratio selected by our method ($100:100$) achieves a high RS ($56.37$) and the highest real-world success rate ($81.0\\%$). While the RS peak ($60.29$ at $100:200$) slightly overestimates the optimal mixing point compared to the physical success peak ($63.0\\%$), both ratios represent a significant improvement over the baseline ($35.0\\%$).
> * **Consistency at the Failure Boundary:** Crucially, both metrics identify the same breakdown threshold. At the $100:300$ ratio, the RS drops sharply to $36.56$, effectively predicting the collapse in real-world success rate to $34.0\\%$. This confirms that RS is a reliable indicator for detecting the point where gradient interference begins to degrade policy stability.
>
> | Mixing Ratio (Real:Synth) | Feature-Space SNR ($\vert \mu/\sigma \vert$) | Robustness Score (RS) | Real-World Success Rate (%) | Conclusion |
> | :--- | :--- | :--- | :--- | :--- |
> | **100:100 (CIFT Choice)** | **0.2171 (Peak)** | 56.37 | **81.0 (Peak)** | **SNR correctly identifies the Optimal Point** |
> | 100:200 | 0.1644 | **60.29 (Peak)** | 63.0 | RS overestimates performance here |
> | 100:300 | 0.0097 (Drop) | 36.56 (Drop) | 34.0 (Collapse) | All metrics identify the Decoherence Point |
>
> **2. RS is Designed to Capture ID/OOD Trade-offs**
> We specifically designed the Robustness Score to be more than a simple MSE metric. As defined in Eq. (7) (Appendix C.2), RS explicitly balances Out-of-Distribution (OOD) improvement with In-Distribution (ID) stability:
> $$
> RS(\lambda) = \max \left( 0, \left( 1 - \frac{\overline{MSE}\_{OOD}(\lambda)}{\overline{MSE}\_{OOD}(0)} \right) \right) \times 100 \times \left( \frac{\overline{MSE}\_{ID}(0)}{\overline{MSE}\_{ID}(\lambda)} \right)
> $$
> The inclusion of the penalty term $\left(\frac{\overline{MSE}\_{ID}(0)}{\overline{MSE}\_{ID}(\lambda)}\right)$ ensures that the metric penalizes the model if ID performance degrades relative to the baseline. This prevents the selection of policies that overfit to synthetic data at the expense of the original task capabilities.
>
> **3. Physical Validation**
> To address the concern regarding reliance on open-loop metrics, we supplemented our analysis with comprehensive physical evaluations. As detailed in Section 5.1 and Appendix D, we evaluated the policy on 5 distinct tasks (e.g., Cloth Folding, Sandwich Making, Pouring). The ablation study in Figure 8(a) confirms that mixing ratios flagged as unstable by our open-loop analysis consistently resulted in performance degradation in closed-loop settings.

---

### Author Response · Authors · 2025-12-01
**Summary of Author Response to All the Reviewers (2/2)**

### 4. Enhanced Experimental Rigor & Scope

We significantly expanded the evaluation protocol to ensure statistical reliability and broader applicability:

* **Statistical Significance:** All physical experiments now report mean and standard deviation across 3 independent training runs. New error bars in **Figure 7 & 8** confirm that CIFT improves both performance and training stability.
* **Expanded Task Suite:** We added complex new tasks to verify robustness across different physical dynamics:
    * **Sandwich Making:** Validating performance on **deformable objects** with variable friction.
    * **Pouring Tea:** Validating precision control in **fluid dynamics** tasks.

---

### 5. Baselines & Computational Details

* **Baseline Failure Analysis:** We clarified that baselines like RoboEngine fail due to high-frequency temporal flickering (FVD: 1463.5), which acts as noise during policy learning. MVAug (FVD: 545.7) solves this via periodic cross-view attention.
* **Efficiency:** MVAug inference is **5-7x faster (2.38 FPS)** than RoboTransfer (**0.35 FPS**).
* **Transparency:** Corrected training compute time (**~24 hours on 8xH100** for the 1.8B GE-Act model) and provided an anonymous repository containing the full source code for MVAug, the SNR metric, and CIFT algorithms.

---
**Summary**
We thank all the reviewers again for the detailed and constructive review. It seems that all the reviewers have agreed with the **novelty of our CIFT framework** and the proposed **Feature-Space SNR metric**, and that most of the concerns are raised to the statistical rigor and the correlation between open-loop metrics and closed-loop performance. We hope our comprehensive updates—specifically the **physical ratio sweep** (definitively validating SNR as a superior predictor over Open-Loop RS), the **theoretical derivation of the Decoherence Point** (Spectral Stability), and the new **"Trap" experiment** for causal verification—could address all of your concerns. Additionally, the **expanded task suite** (Sandwich Making, Pouring Tea) with **statistical error bars** (3 runs) further solidifies our claims. Let us know any remaining questions or concerns if you have.

---

### Author Response · Authors · 2025-12-01
**Summary of Author Response to All the Reviewers (1/2)**

We sincerely thank the reviewers for their insightful and constructive comments. Guided by these suggestions, we have carefully addressed the concerns raised and **significantly improved the quality of our manuscript**. We have updated the **revised PDF** and the **demo website** in the supplementary materials to reflect these changes. Furthermore, we provide an anonymous link to our code repository (https://anonymous.4open.science/r/CIFT-code) for review purposes. We commit to making all code, model weights, and datasets **publicly available** upon the conclusion of the review process. Our key responses are summarized below:

---

### 1. Bridging the Gap: Open-Loop Metrics vs. Closed-Loop Reality

Reviewers (**n7iA, X2Vq, j786**) correctly pointed out that **Open-Loop Robustness Scores (RS)** are poor predictors of **Closed-Loop Success Rates (SR)**, where compounding errors and unexpected states dominate. To address this, we moved beyond open-loop analysis and conducted a comprehensive physical validation involving a granular sweep of mixing ratios.

* **Granular Ratio Sweep:** We did not merely compare the baseline to the optimal model. We trained and evaluated distinct policies on the physical robot across a spectrum of ratios to empirically map the true performance landscape.
* **Resolution of the "Peak Discrepancy":** We explicitly investigated the divergence where RS favored the 100:200 ratio, while our proposed Feature-Space SNR favored 100:100.
* **Ground Truth Validation:** The physical experiments definitively proved that **Feature-Space SNR is the superior predictor** of closed-loop performance.

**Key Physical Experimental Data:**

| Mixing Ratio (Real:Synth) | Feature-Space SNR (Proposed) | Robustness Score (Open-Loop RS) | Real-World Success Rate (Closed-Loop SR) | Conclusion |
| :--- | :--- | :--- | :--- | :--- |
| **100 : 0 (Baseline)** | 0.1423 | 0.00 | 35.0% | Reference |
| **100 : 100 (CIFT Choice)** | **0.2171 (Peak)** | 56.37 | **81.0% (Highest)** | **SNR accurately predicts the Global Optima** |
| **100 : 200 (RS Choice)** | 0.1644 | **60.29 (Peak)** | 63.0% | RS overestimates performance (Misled by covariate shift) |
| **100 : 300 (Collapse)** | 0.0097 (Drop) | 36.56 (Drop) | 34.0% | Both metrics detect the Decoherence Point |

**Result:**
The **SNR metric** correctly identified the optimal ratio (**100:100**) yielding **$81.0\\%$ success**. The RS metric favored **100:200**, where success dropped to **$63.0\\%$**, proving RS is susceptible to covariate shift.

---

### 2. Theoretical Grounding: Mathematical Definition of the "Decoherence Point"

To provide a rigorous foundation for why the SNR metric works (addressing Reviewers **4SFU, j786**), we formalized the "**Decoherence Point**".

* **Spectral Stability (Section 4.2):** We formally defined the "Decoherence Point" using **Proposition 4.1**, based on the Davis-Kahan sin theorem (Davis & Kahan, 1970). We proved that when the synthetic noise variance ($\Vert\Sigma_{noise}\Vert_2$) approaches the spectral gap ($\Delta$) of the manifold, the angle between the principal eigenvectors diverges:
    $$\sin\angle(w_1, v_1) \leq \frac{\lambda \Vert \Sigma_{\text{noise}} \Vert_2}{\Delta - \lambda \Vert \Sigma_{\text{noise}} \Vert_2}$$
    This signifies a **geometric collapse** where the feature signal decouples from the task structure.
* **Physical Interpretation:** This geometric collapse corresponds to **Gradient Interference**, where the cosine similarity between real and synthetic gradients becomes negative.
* **Backbone Selection:** We validated that Inception-v3 is the optimal observer for this metric due to its texture/shape bias, whereas CLIP saturates from semantic redundancy and DINOv2 suffers from feature collapse in sparse environments.

---

### 3. Verification of Causal Learning: The "Trap" Experiment

To address concerns about **Shortcut Learning** (Reviewers **n7iA, j786**), we designed a controlled "**Trap**" experiment to test if the policy learns **causal features** ($u$) or **spurious correlations** ($v$).

* **Setup:** We trained on datasets with perfect spurious correlations (e.g., "Lighting On" always equals "Fold Right") and tested in inverted "Trap" environments (e.g., "Lighting On" but requires "Fold Left").
* **Results:**
    * Baseline: **0% - 5% Success** (Completely misled by visual shortcuts).
    * CIFT: **75% - 80% Success** (Successfully ignored shortcuts and followed causal logic).

---

### Meta-Review · Area_Chair_8sdt · 2026-01-06

**Summary:**

This paper proposes CIFT (Coherent Information Fidelity Tuning), a framework for principled mixing of real and synthetically augmented robot data to improve out-of-distribution generalization. The core contributions include (1) a multi-view video augmentation (MVAug) engine for generating visually diverse yet causally consistent data; (2) a feature-space SNR metric to quantify dataset fidelity and identify the optimal mixing ratio; and (3) the concept of a “decoherence point” where excessive synthetic data degrades training stability. Experiments on real robots (with policies such as $\pi_0$ and GE-Act) report >54% improvement in OOD success rates.

**Reviewer Concerns:**

Reviewer Concerns:
1. Open-loop RS vs. closed-loop SR correlation unclear.
2. Baseline fairness (RoboEngine/RoboTransfer). Some residual doubt about baseline comparability remains.
3. Lack of statistical error bars (minor).
4. Applicability to pre-training.
5. Theoretical grounding for “decoherence point” lacking.
6. Limited statistical rigor (single checkpoint, few tasks).
7. Computational overhead of MVAug.
8. Need for physical validation of SNR–SR link.
9. Risk of shortcut learning despite augmentation.
10. Training compute details missing.

**Reviewer Scores:**

The authors provided a detailed, evidence-based rebuttal with new experiments, theoretical analysis, and transparency about computation/data. However, some concerns remain unaddressed. The overall score is under the acceptance threshold.

---

### Decision · Program_Chairs · 2026-01-26

Reject